# SOX4 facilitates PGR protein stability and FOXO1 expression conducive for human endometrial decidualization

Pinxiu Huang[1,2,3,4†], Wenbo Deng[2†], Haili Bao[2], Zhong Lin[3], Mengying Liu[2], Jinxiang Wu[2], Xiaobo Zhou[2], Manting Qiao[2], Yihua Yang[1], Han Cai[2], Faiza Rao[2], Jingsi Chen[5], Dunjin Chen[5], Jinhua Lu[2], Haibin Wang[2*], Aiping Qin[1*], Shuangbo Kong[2*]

[1]Department of Reproductive Medicine, The First Affiliated Hospital of Guangxi Medical University, Nanning, China; [2]Fujian Provincial Key Laboratory of Reproductive Health Research, Department of Obstetrics and Gynecology, The First Affiliated Hospital of Xiamen University, School of Medicine, Xiamen University, Xiamen, China; [3]Department of Reproductive Medicine, Liuzhou Maternity and Child Health Hospital, Liuzhou, China; [4]Affiliated Maternity Hospital and Affiliated Children's Hospital of Guangxi University of Science and Technology, Liuzhou, China; [5]Department of Obstetrics and Gynecology, Key Laboratory for Major Obstetric Diseases of Guangdong Province, The Third Affiliated Hospital of Guangzhou Medical University, Guangzhou, China

**\*For correspondence:**
haibin.wang@vip.163.com (HW);
qinaiping@gxmu.edu.cn (AQ);
shuangbo_kong@163.com (SK)

†These authors contributed equally to this work

**Competing interest:** The authors declare that no competing interests exist.

## Abstract

The establishment of pregnancy in human necessitates appropriate decidualization of stromal cells, which involves steroids regulated periodic transformation of endometrial stromal cells during the menstrual cycle. However, the potential molecular regulatory mechanism underlying the initiation and maintenance of decidualization in humans is yet to be fully elucidated. In this investigation, we document that SOX4 is a key regulator of human endometrial stromal cells decidualization by directly regulating FOXO1 expression as revealed by whole genomic binding of SOX4 assay and RNA sequencing. Besides, our immunoprecipitation and mass spectrometry results unravel that SOX4 modulates progesterone receptor (PGR) stability through repressing E3 ubiquitin ligase HERC4-mediated degradation. More importantly, we provide evidence that dysregulated SOX4–HERC4–PGR axis is a potential cause of defective decidualization and recurrent implantation failure in in-vitro fertilization (IVF) patients. In summary, this study evidences that SOX4 is a new and critical regulator for human endometrial decidualization, and provides insightful information for the pathology of decidualization-related infertility and will pave the way for pregnancy improvement.

## Editor's evaluation

The manuscript provides a novel mechanism of progesterone receptor stability mediated by SOX4 in human endometrial decidualization. The authors have addressed all the concerns raised by the reviewers and in addition provided the gels at better resolution, leading to a significantly improved paper.

## Introduction

Adequate crosstalk between implantation-competent embryo and receptive endometrium is prerequisite for successful pregnancy (*Cha et al., 2012*). The receptive endometrium in human requires remodeling of stromal cells under the regulation of rising progesterone and intracellular cyclic AMP to initiate the decidualization, which will undergo more extensive transformation after embryo implantation (*Gellersen and Brosens, 2014*). The human endometrial stromal compartment can decidualize during the progesterone-dominant early secretory phase of a nonconception cycle (*Wang et al., 2020*). During the pregnancy, the decidual process is poised to transit through distinct phenotypic phases that underpin endometrial receptivity, embryo selection, and ultimately resolution of pregnancy (*Gellersen and Brosens, 2014*). The decidual reaction plays a central role in the establishment of a pregnancy and continues throughout the pregnancy. Insufficient decidualization in endometrium is related to failed embryo implantation (*Peter Durairaj et al., 2017*), unexplained infertility, recurrent spontaneous abortion (*Coulam, 2016*), intrauterine growth retardation (*Lefèvre et al., 2011*), and preeclampsia (*Garrido-Gomez et al., 2017*). However, the underlying molecular mechanism governing the endometrial decidualization remains enigmatic (*Okada et al., 2018*).

A wide range of transcription factors crucial for stromal cell decidualization have been successively identified (*Gellersen and Brosens, 2014*). FOXO1, a member of FOXO fork-head transcription factors subfamily, is one of the earliest identified transcriptional factors in human endometrial stromal cells (HESCs) responding to decidualization stimulation (progesterone and cAMP) (*Labied et al., 2006*). Accumulative evidence has demonstrated that FOXO1 regulates transcription of PRL and IGFBP1 through direct binding to their promoters (*Christian et al., 2002*; *Kim et al., 2005*). Progesterone receptor (PGR), which is a critical master factors for the endometrial stromal cells (ESCs) decidualization, imparts endometrium receptivity through binding with P4 ligand and its nuclear translocation (*Keller et al., 1979*; *Mulac-Jericevic et al., 2000*). An array of PGR direct target genes had been unraveled by PGR ChIP-Seq in both human and mice (*Rubel et al., 2012*; *Mazur et al., 2015*; *Chi et al., 2020*). Abnormal PGR expression is closely relevant with unexplained infertility (*Keller et al., 1979*) and endometriosis (EMS) (*Zhou et al., 2016*; *Pei et al., 2018*). Although many transcription factors and downstream events in decidualization of both mice and human have been unraveled (*DeMayo and Lydon, 2020*), the precise mechanism orchestrating transcriptional regulatory network underpinning endometrial decidualization remained not fully explored.

SOX4 is a highly conserved transcription factor belonging to the SOX (SRY-box) family. Studies have shown that SOX4 is vital to a variety of biological processes, including embryogenesis, neural development, and differentiation (*Moreno, 2020*). Further, it has been noticed that SOX4 knockout mice die of cardiac malformation on the 14th day of pregnancy, suggesting the key role of SOX4 in embryonic development (*Ya et al., 1998*). In addition, an increasing number of reports indicate that SOX4 is related to tumor cell proliferation, metastasis, and epithelial–mesenchymal transformation (*Li et al., 2020*). Report also shows that SOX4 is highly expressed in breast cancer under the regulation of progesterone (*Graham et al., 1999*). Additionally, SOX17, another member of the SOX family, has also been observed to be a direct target of PGR in epithelium regulating IHH expression (*Wang et al., 2018*). While the significance and regulation of SOX4 in female pregnancy remain intangible.

Here, we provide evidence that SOX4, under the regulation of P4-PGR, guides human endometrium stromal cells (HESCs) decidualization by regulating FOXO1 expression as revealed by ChIP-Seq and RNA sequencing (RNA-Seq). Mechanism studies also unravel the importance of SOX4 in maintaining the protein stability of PGR by repressing ubiquitin E3 ligase HERC4. Moreover, both SOX4 and PGR have been demonstrated to be aberrantly downregulated in the endometrium of EMS patients suffering from implantation failure.

## Results

### SOX4 is dynamically expressed in human endometrial stromal cells regulated by P4-PGR signaling

To investigate the expression of transcription factors in HESCs, RNA-Seq was performed in normal endometrium stromal cells. We noted that among the transcription factors expressed in endometrium stromal cells, SOX4 was the 17th highest expressed (*Figure 1A*), and as the most abundant member in SOX family (*Figure 1B*). Meanwhile, previous RNA-Seq data in isolated mouse stromal

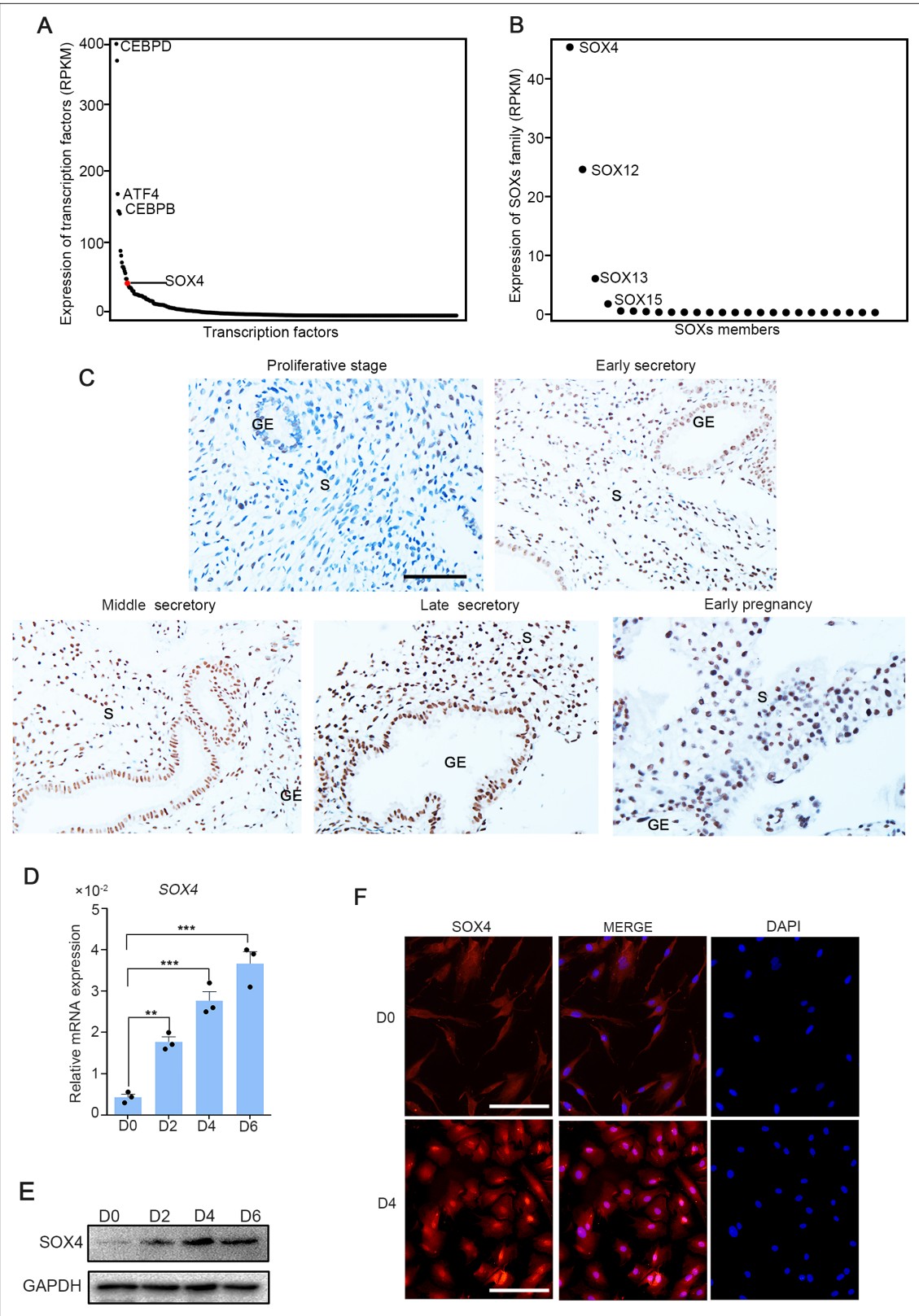

**Figure 1.** SOX4 is dynamically expressed in human ESCs. Expression of all transcription factors (**A**) and SOX family genes (**B**) in human nondecidualized ESCs by RNA-Seq. The value in *Y*-axis indicated the RPKM (reads per kilobase per million mapped reads) in RNA-Seq data. (**C**) Immunohistochemical analysis of endometrial SOX4 protein expression in proliferative, secretory phases (early, middle, and late) of the menstrual cycle and early pregnancy (about 8 weeks). GE: gland epithelium; S: stroma. Scale bar: 100 µm. (**D**) Expression of *SOX4* mRNA levels in decidualized stromal cells at different time

*Figure 1 continued on next page*

*Figure 1 continued*

points after the E2, MPA, and cAMP treatment. Results are presented as means ± standard error of the mean (SEM); $n = 3$; **p < 0.005; ***p < 0.0001. (**E**) Expression of SOX4 protein levels in decidualized stromal cells at different time points after the E2, MPA, and cAMP treatment. (**F**) Immunofluorescent detection of SOX4 protein localization in the undecidualized (D0) and decidualized (D4) human endometrial stromal cells (HESCs). Scale bar: 100 µm.

The online version of this article includes the following figure supplement(s) for figure 1:

**Figure supplement 1.** Sox family expression in mouse uterine stromal and epithelial cells.

**Figure supplement 2.** SOX4 expression in the primary stromal cell during the decidualization.

and epithelial cells revealed that SOX4 expression was the most abundant SOX family in stromal cells (*Figure 1—figure supplement 1A*), while the expression of SOX17 was restricted to the epithelium (*Deng et al., 2019*). Thus, we speculated that the conserved expression of SOX4 in stroma was linked to decidualization. To explore whether SOX4 was under the regulation of dynamic change of estrogen and progesterone during the menstrual cycle, we first analyzed the expression of SOX4 in endometrial biopsy samples obtained from healthy, reproductive-aged volunteers with regular menstrual cycles. Albeit SOX4 was detected at a low level in the E2-dominant proliferative phase, its expression was progressively increased in the nucleus of stromal cellsin P4-dominant early, middle, late secretory phases and this high expression in stroma cell sustained in the decidual cells in early pregnancy (*Figure 1C*).

This upregulation of SOX4 in differentiated stromal cell in secretory phase was also substantiated in in vitro decidualized stromal cells induced by E2, MPA, and cAMP (EPC) cocktail, displaying gradually cumulated mRNA and protein levels in immortalized HESCs (*Figure 1D, E*) as well as in primary

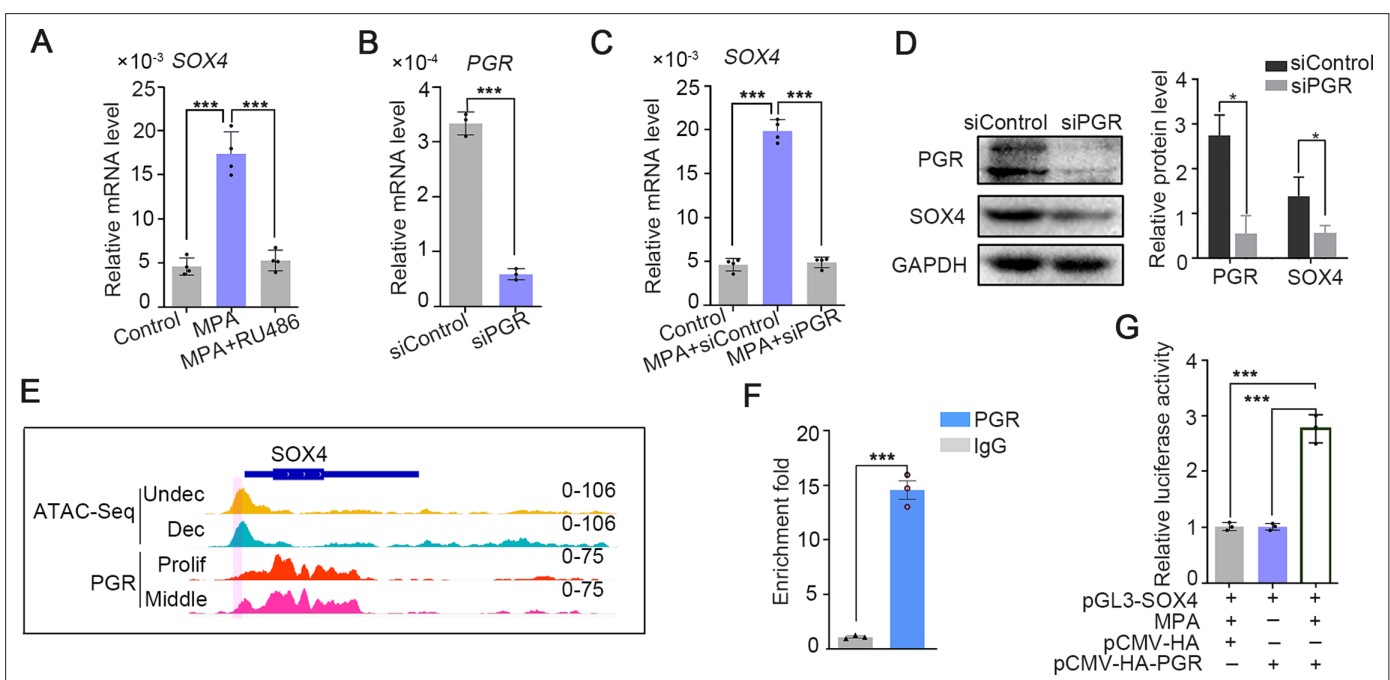

**Figure 2.** SOX4 is regulated by P4-progesterone receptor (PGR) signaling in human ESCs. (**A**) Expression of *SOX4* mRNA in the presence of MPA or MPA + RU486 for 2 days in immortalized human endometrial stromal cells (HESCs). Results are presented as means ± standard error of the mean (SEM); $n = 3$; ***p < 0.0001. (**B**) RNA level of PGR after siRNA-mediated knockdown. Results are presented as means ± SEM; $n = 3$; ***p < 0.0001. (**C**) Expression of *SOX4* mRNA in the presence of MPA for 2 days with PGR knockdown. Results were presented as means ± SEM; $n = 3$; ***p < 0.0001. (**D**) Protein level of SOX4 and PGR after PGR knockdown in the presence of MPA for 2 days. Band quantification of PGR and SOX4 protein, relative to loading control GAPDH. *p < 0.05 ($n = 3$). (**E**) Visualization of PGR binding and chromatin accessibility on SOX4 locus. The chromatin accessibility is depicted in undecidualized and decidualized stromal cells and genome-wide PGR binding is generated from proliferated and middle secretory endometrium as revealed from previous reports. Undec: undecidualized HESCs; Dec: decidualized HESCs; Prolif: proliferative endometrium; Middle: middle phase of secretory endometrium. (**F**) ChIP assay of potential PGR binding on SOX4 as indicated from (**E**) in decidualized immortalized HESCs for 2 days. Data are plotted as mean ± SEM; $n = 3$; ***p < 0.0001. (**G**) Luciferase activity assay of SOX4 promotor in the presence of MPA and PGR in 293T cells. Results are presented as means ± SEM. $n = 3$; *p < 0.05; **p < 0.005; ***p < 0.0001.

stromal cells (*Figure 1—figure supplement 2A, B*). Immunofluorescence also manifested that SOX4 was mainly localized in the nucleus of HESCs after EPC treatment (*Figure 1F*).

The intense expression of SOX4 in the stroma of secretory phase accompanied with rising P4 and inducted by EPC in in vitro stromal cells, suggesting that progesterone may be involved in the regulation of SOX4 expression. This assumption was supported by the observation that SOX4 expression was induced by 2 days treatment of MPA, a progesterone analog commonly used in decidualization induction, and abrogated in the presence of PGR antagonist RU486 (*Figure 2A*). Progesterone-induced SOX4 mRNA and protein levels were significantly attenuated with PGR knockdown in immortalized HESCs (*Figure 2B–D*). Thus, the P4-PGR signaling was critical for SOX4 expression in HESCs. Based on previous ATAC-Seq data from undecidualized and decidualized stromal cells and PGR ChIP-Seq from proliferative and secretory endometrium (*Chi et al., 2020*), we found that there was intensive potential PGR binding at SOX4 promoter (*Figure 2E*). This speculation was validated by PGR ChIP-qPCR at SOX4 promoter in decidualized stromal cells (*Figure 2F*). Furthermore, the binding of PGR on SOX4 promoter was also confirmed by luciferase reporter assay that overexpression of PGR increased the reporter activities in the presence of MPA in 293T cells (*Figure 2G*). These results indicated that PGR guided SOX4 expression via direct transcriptional regulation.

## SOX4 is required for human endometrial stromal cell decidualization

To depict the significance of SOX4 during decidualization, SOX4 was knockout by CRISPR/Cas9 which was ascertained by Sanger-sequencing in both alleles (*Figure 3—figure supplement 1A*) and further verified by western blot and immunofluorescence (*Figure 3—figure supplement 1B, C*). The expression of PRL and IGFBP1 was dramatically decreased in SOX4 knockout HESCs after decidualization for 2, 4, and 6 days compared with SOX4 intact cells (*Figure 3A–C*). Similar results were obtained in primary HESC after SOX4 knockdown by shRNA prior to EPC administration (*Figure 3—figure supplement 2A–D*). On the other side, overexpressing SOX4 in HESCs significantly augmented the expression of *PRL* and *IGFBP1* in both immortalized (*Figure 3D–F*) and primary HESC (*Figure 3—figure supplement 2A–C*) decidualized for 4 days.

Moreover, RNA-Seq was utilized in SOX4 intact or depleted immortalized HESCs treated with EPC for 2 days. The genes significantly downregulated after SOX4 knockdown included aforementioned *IGFBP1* and *PRL* as well as other genes critical for decidualization, such as *FOXO1*, *LEFTY2*, *WNT5A*, *WNT4*, *FOSL2*, *STAT3*, *LEFTY2*, *IL-11*, and *BMP2* (*Figure 3G, H*). Kyoto Encyclopedia of Genes and Genomes (KEGG) analysis showed that the differentially expressed genes were enriched in the FOXO signaling pathway, TGF-beta signaling pathway, and MAPK signaling pathway (*Figure 3I*), consistent with the observed defective decidualization in the absence of SOX4. The Gene Ontology (GO) annotation analysis showed that the differentially expressed genes were related to autophagy-associated pathway, insulin-like growth factor binding, collagen-containing extracellular matrix, and cell–substrate adhesion (*Figure 3J*). Together, these results revealed the critical role of SOX4 in HESCs decidualization.

## Genome binding of SOX4 in decidualized stromal cells

To interrogate genes directly regulated by SOX4, we detected the chromatin-wide binding of SOX4 by ChIP-Seq. Our results showed that there were total of 23,709 SOX4-binding peaks with most of them enriched in promoters, distal intergenic and intron (*Figure 4A*). Peak enrichment indicated that SOX4 binding was primary closed to TSS as well as evidenced by heatmap of peak distribution (*Figure 4B, C*). We also noticed that SOX4 mainly binds to those genes with higher expression (RPKM ≥1) accompanied with few SOX4 binding on those less to no expression genes (RPKM <1) (*Figure 4B*). Motif analysis results revealed that the most significantly enriched motif was SOX4 consensus binding sites, further supported the reliability of this SOX4 ChIP-Seq. We also noticed that FRA2 and FOSL2, two critical members of AP1, were also highly enriched at SOX4-binding sites, indicating the potential cooperation between SOX4 and AP1 (*Figure 4D*). Considering the correlation between SOX4 binding and gene expression, we overlapped SOX4-binding genes with those downregulated genes after SOX4 knockdown and found that 389 genes with SOX4 binding decreased after SOX4 ablation, indicating the direct regulation of SOX4 on these genes (*Figure 4E*).

It was gratifying that there were several latent SOX4-binding sites on the STAT3, FOXO1, and FOSL2 in both stromal cell line and primary cultured stromal cells (*Figure 4F*). ChIP-qPCR also

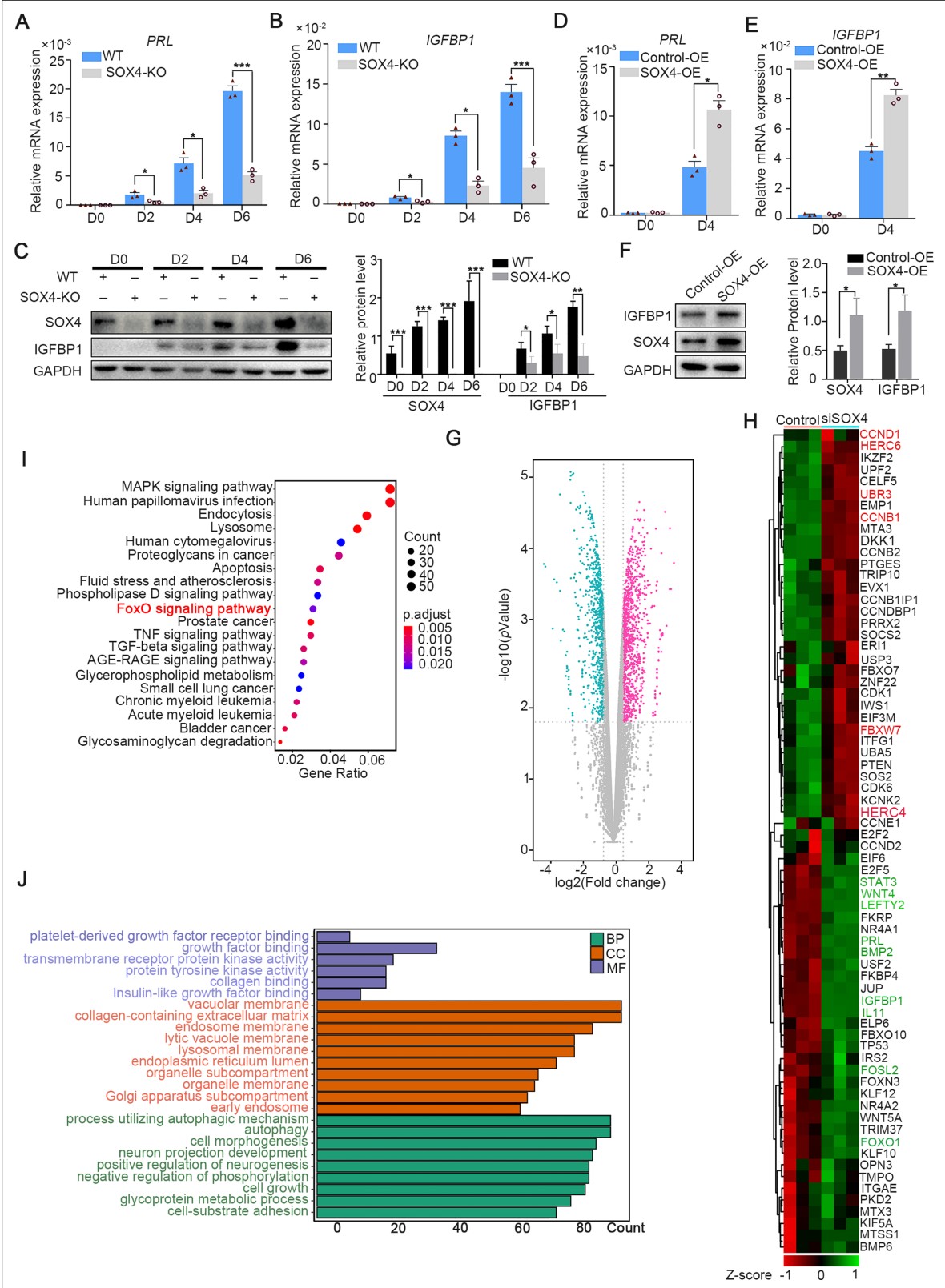

**Figure 3.** SOX4 regulated genes in decidualized human endometrial stromal cells (HESCs). mRNA levels of *PRL* (A) and *IGFBP1* (B) in undecidualized and decidualized immortalized HESCs after SOX4 knockout at indicated time points after the E2, MPA, and cAMP treatment. Results are presented as means ± standard error of the mean (SEM); *n* = 3; *p < 0.05; ***p < 0.0001. (**C**) Protein levels of IGFBP1 in undecidualized and decidualized HESCs after SOX4 knockout at indicated time points. (**D, E**) mRNA levels of *PRL* (**C**) and *IGFBP1* (**D**) after SOX4 overexpression in undecidualized or decidualized

*Figure 3 continued on next page*

*Figure 3 continued*

HESCs for 4 days. *n* = 3; *p < 0.05. (**F**) Protein levels of SOX4 and IGFBP1 in decidualized HESCs for 4 days with control or SOX4 overexpression. Band quantification of SOX4 and IGFBP1 protein, relative to GAPDH. (**G**) Differential expressed genes detected by RNA sequencing (RNA-Seq) in immortalized HESCs decidualized for 2 days with control or SOX4 knockdown as visualized by volcano plot. (**H**) Heatmap of top differential expressed genes from RNA-Seq after SOX4 knockdown in decidualized HESCs. (**I, J**) KEGG (**J**) and Gene Ontology (GO) (**K**) analysis of the differentially expressed genes in RNA-Seq. Results were presented as means ± standard error of the mean (SEM). *p < 0.05; **p < 0.005; ***p < 0.0001. The above experiments were repeated three times.

The online version of this article includes the following figure supplement(s) for figure 3:

**Figure supplement 1.** SOX4 knockout in the stromal cell by CRISPR–Cas9.

**Figure supplement 2.** SOX4 knockdown by the shRNA derailed the decidualization in primary stromal cell.

**Figure supplement 3.** SOX4 overexpression upregulated the expression of decidualization marker genes.

confirmed the binding of SOX4 on these genes (*Figure 4G–J*). Those downregulated genes (900) after SOX4 knockdown possess more SOX4-binding reads compared with upregulated genes (1022) (*Figure 4K*), indicating the transcriptionalactivation effect of SOX4. Importantly, those SOX4 direct target genes also showed specific enrichment of the FOXO signaling pathway (*Figure 4L*). To further assess the regulation of SOX4 on FOXO1, SOX4 was knockout and the expression of FOXO1 was overtly decreased accompanied with compromised decidualization as marked by lessened IGFBP1 during the decidualization from days 2 to 6 (*Figure 4M, N*). On the other hand, FOXO1 mRNA and protein were upregulated in SOX4 overexpression stromal cell decidualized for 4 days, as well as IGFBP1 protein (*Figure 4O*, and *Figure 4—figure supplement 1B*). Moreover, the effects of SOX4 overexpression on decidualization were blocked when the FOXO1 was knockdown, suggested the FOXO1 as a crucial effector downstream of SOX4 (*Figure 4—figure supplement 1A, B*). Besides the downregulated genes, there were also 438 genes with SOX4-binding sites were upregulated and GO analysis revealed that cell cycle regulators were mostly enriched (*Figure 4—figure supplement 2A, B*). During the decidual induction, the stroma cell would stop proliferation and initiate the differentiation program, the aberrant cell mitotic division in the absence of SOX4 was consistent with the defective decidualization. In summary, these results suggested that SOX4 directly regulated transcription of transcription factor FOXO1, as well as other molecules conducive for human endometrial decidualization.

## SOX4 stabilizes PGR through repressing ubiquitin–proteasome pathway

Previous studies showed that P4-PGR signaling was critical for both initiation and maintenance of decidualization (*Gellersen and Brosens, 2014*), which incited us to assess PGR expression in the absence of SOX4 during the decidualization. We were attracted by the finding that PGR protein levels, including both the PRA and PRB isoforms, were dramatically decreased in the absence of SOX4 on days 0, 2, 3, and 6 during the decidualization without overt transcription reduction (*Figure 5A, B*). Similar results were also observed in primary stromal cells, implying that SOX4 stabled PGR protein at post-transcriptional level (*Figure 5—figure supplement 1A, B*). Furthermore, treatment of cells with the proteasome inhibitor MG-132 for 6 hr efficiently attenuated the rapid degradation of PGR protein in SOX4-deficient cells (*Figure 5C*), but the autophagy inhibitor 3-methyladenine (3-MA) cannot rescue or attenuate the degradation of PGR protein (*Figure 5—figure supplement 1C*), suggested the PGR protein was mainly degraded through the proteasome in the absence of SOX4. Likewise, PGR protein degeneration was faster in SOX4-depleted HESCs 2 days after EPC treatment in the presence of the protein synthesis inhibitor cycloheximide (CHX) for 0, 3, 6, and 9 hr (*Figure 5D, E*). Thus, these results revealed that SOX4 knockdown led to shortened the half-life of PGR protein.

To further explore the regulatory apparatus of PGR protein stability, the ubiquitination status of PGR was determined. Immunoprecipitated PGR was blotted against ubiquitin antibody in decidualized immortalized HESCs pretreated with MG-132 for 6 hr to postpone protein degradation. Notably, PGR ubiquitination was increased after SOX4 depletion (*Figure 5F*). These results suggested that SOX4 knockdown reduced PGR protein stability through the polyubiquitin-mediated proteasome degradation pathway.

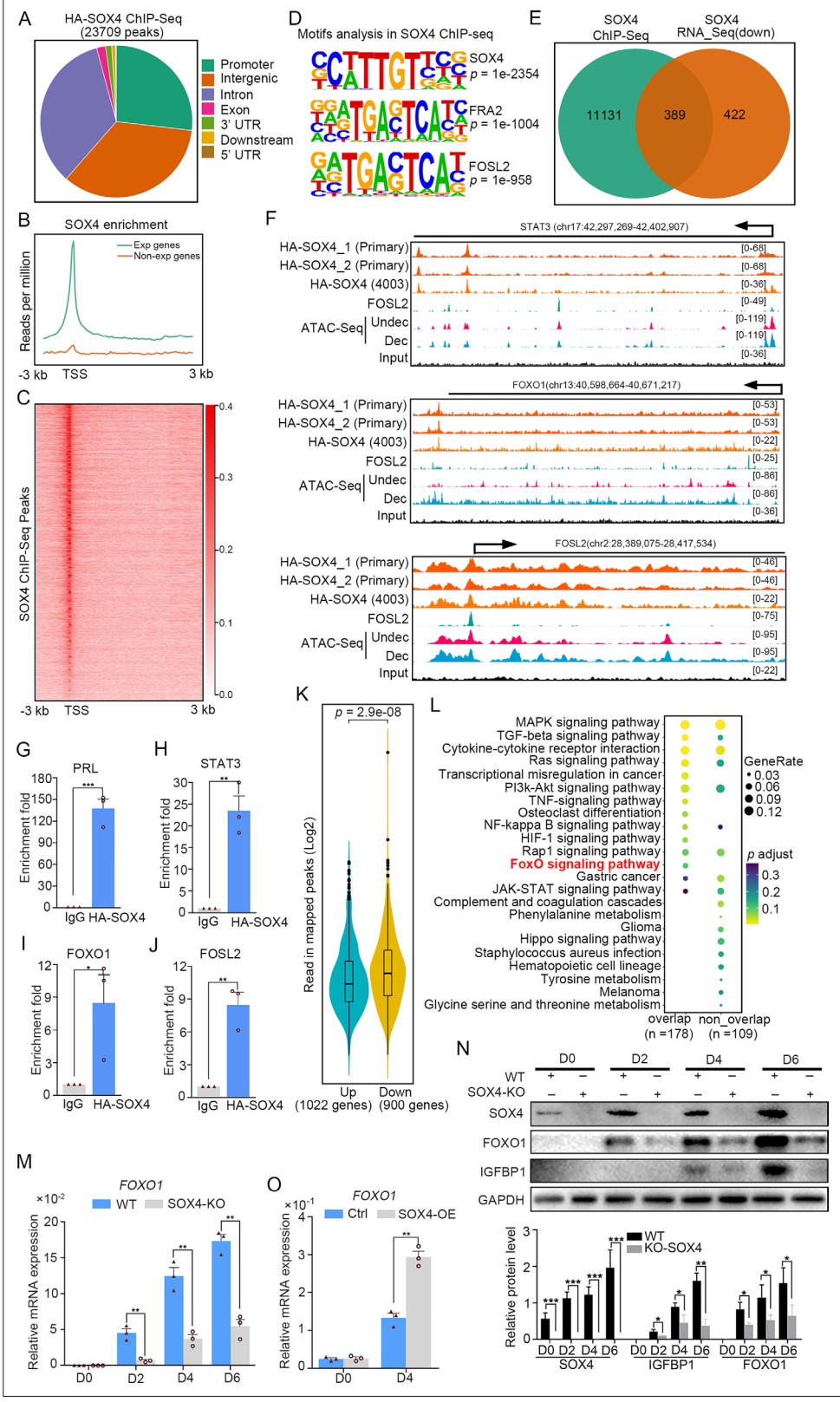

**Figure 4.** Genome wide binding of SOX4 in decidualized stromal cells. (**A**) Distribution of SOX4-binding peaks as revealed from SOX4 ChIP-Seq in SOX4 overexpressed (HA-SOX4) immortalized human endometrial stromal cells (HESCs) decidualized for 2 days. (**B**) Distribution SOX4 binding in genebody. (**C**) Heatmap of SOX4-binding sites distribution. (**D**) Motif analysis of SOX4-binding sites. (**E**) Venn diagram of SOX4 directly binding peaks and

*Figure 4 continued*

SOX4 regulated genes after SOX4 knockdown. (**F**) SOX4-binding site in the STAT3, FOXO1, and FOSL2 with progesterone receptor (PGR) binding and chromatin accessibility in both primary stromal cell and stromal cell line. The chromatin accessibility is depicted in undecidualized and decidualized stromal cells and genome-wide PGR binding is generated from proliferated and middle secretory endometrium as revealed from previous reports. Undec: undecidualized HESCs; Dec: decidualized HESCs. (**G–J**) ChIP-qPCR assay of SOX4 binding on PRL, STAT3, FOXO1, and FOSL2 in HA-SOX4 overexpressed decidualized HESCs. Results were presented as means ± standard error of the mean (SEM); $n$ = 3; *$p < 0.05$; **$p < 0.005$; ***$p < 0.0001$. (**K**) Read number of SOX4 binding in SOX4 regulated genes after SOX4 knockdown. (**L**) KEGG analysis of overlapped genes of SOX4 directly binding and SOX4 downregulated genes as well as nonoverlapped genes. (**M**) mRNA levels of *FOXO1* in SOX4 knockout undecidualized and decidualized HESCs at indicated time points. Results were presented as means ± SEM; $n$ = 3; **$p < 0.005$. (**N**) Protein levels of SOX4, FOXO1, and IGFBP1 in SOX4 knockout undecidualized and decidualized HESCs at indicated time points. Band quantification of indicated proteins, relative to loading control GAPDH. *$p < 0.05$; ***$p < 0.0001$. $n$ = 3. (**O**) mRNA levels of *FOXO1* after SOX4 overexpression in undecidualized or decidualized HESCs for 4 days. Results were presented as means ± SEM; $n$ = 3; **$p < 0.005$. The above experiments were repeated three times.

The online version of this article includes the following figure supplement(s) for figure 4:

**Figure supplement 1.** FOXO1 knockdown abolished the effect of SOX4 overexpression on decidualization.

**Figure supplement 2.** The upregulated genes after SOX4 knockdown with SOX4-binding sites in their regulatory regions.

## SOX4 inhibits PGR degradation by regulating E3 ligase HERC4

Since ubiquitin E3 ligases were mainly responsible for protein ubiquitination (**Rape, 2018**), we next intended to identify the potential ubiquitin E3 ligase for PGR protein. Mass spectrometry (MS) was employed for PGR immunoprecipitated (IP) proteins in decidualized immortalized HESCs to estimate PGR-associated proteins (**Figure 6—figure supplement 1A**). Reassuringly, several E3 ubiquitination ligases were identified, including HERC4, RNF213, and UBR3 whose expressions were also upregulated according to RNA-Seq in SOX4-silenced HESCs (**Figure 6A**). Real-time PCR confirmed the abnormal upregulation of these E3 ligases when SOX4 was downregulated (**Figure 6B**, and **Figure 6—figure supplement 1B, C**). The protein level of HERC4 was consistently upregulated upon the ablation of SOX4 (**Figure 6C**). As HERC4 showed the most significant change after SOX4 knockdown and its interaction with PGR as a potential E3 ubiquitin ligase was also supported by UbiBrowser database (http://ubibrowser.ncpsb.org), we mainly focused on this E3 ligase in our following experiments.

To verify whether HERC4 was a potential E3 ligase for PGR protein, we first detected the protein interaction between HERC4 and PGR. The coimmunoprecipitation result demonstrated a conserved physical interaction between endogenous HERC4 and PGR in decidualized stromal cells as well asexogenous expressed in 293T cells (**Figure 6D, E**). Immunofluorescence analysis showed the colocalization of HERC4 and PGR in decidualized HESCs (**Figure 6F**). To figure out whether HERC4 could induce endogenous ubiquitination of PGR, HERC4 was overexpressed in immortalized HESCs followed by decidualization for 2 days. The ubiquitination of PGR was higher in the presence of HERC4 than in control, as shown in **Figure 6G**, consistent with a reduced level of PGR protein in immortalized HESCs cells (**Figure 6—figure supplement 1D**). Conversely, HERC4 abolished by siRNA significantly reduced PGR ubiquitination in 293T cells (**Figure 6H**). The above studies demonstrated that HERC4 was a critical E3 ligase for PGR ubiquitination. Moreover, overexpression of HERC4 recused the decreasing PGR ubiquitination in the presence of SOX4 in 293T cells (**Figure 6I**). Hence, SOX4 affected PGR ubiquitination through E3 ubiquitin ligase HERC4.

Since HERC4 was a ubiquitin ligase of PGR and the increased ubiquitination of PGR was observed in the presence of HERC4, we next interrogated whether HERC4-mediated PGR degradation. This posit was underpinned by decreased PGR protein level with HERC4 overexpression in both 293T cells (**Figure 6J**) and immortalized HESCs (**Figure 6K**). PGR protein was also rescued by siRNA-mediated knockdown of HERC4 (**Figure 6—figure supplement 1E**) in SOX4 abolished decidualized immortalized HESCs (**Figure 6L**), and consistently, the IGFBP1 level was also rescued (**Figure 6—figure supplement 1F**). Likewise, PGR half-life increased in both normal and SOX4 abolished immortalized HESCs after HERC4 repression (**Figure 6M, N**). In a word, these studies demonstrated that SOX4-mediated PGR degradation by modulating E3 ligase HERC4.

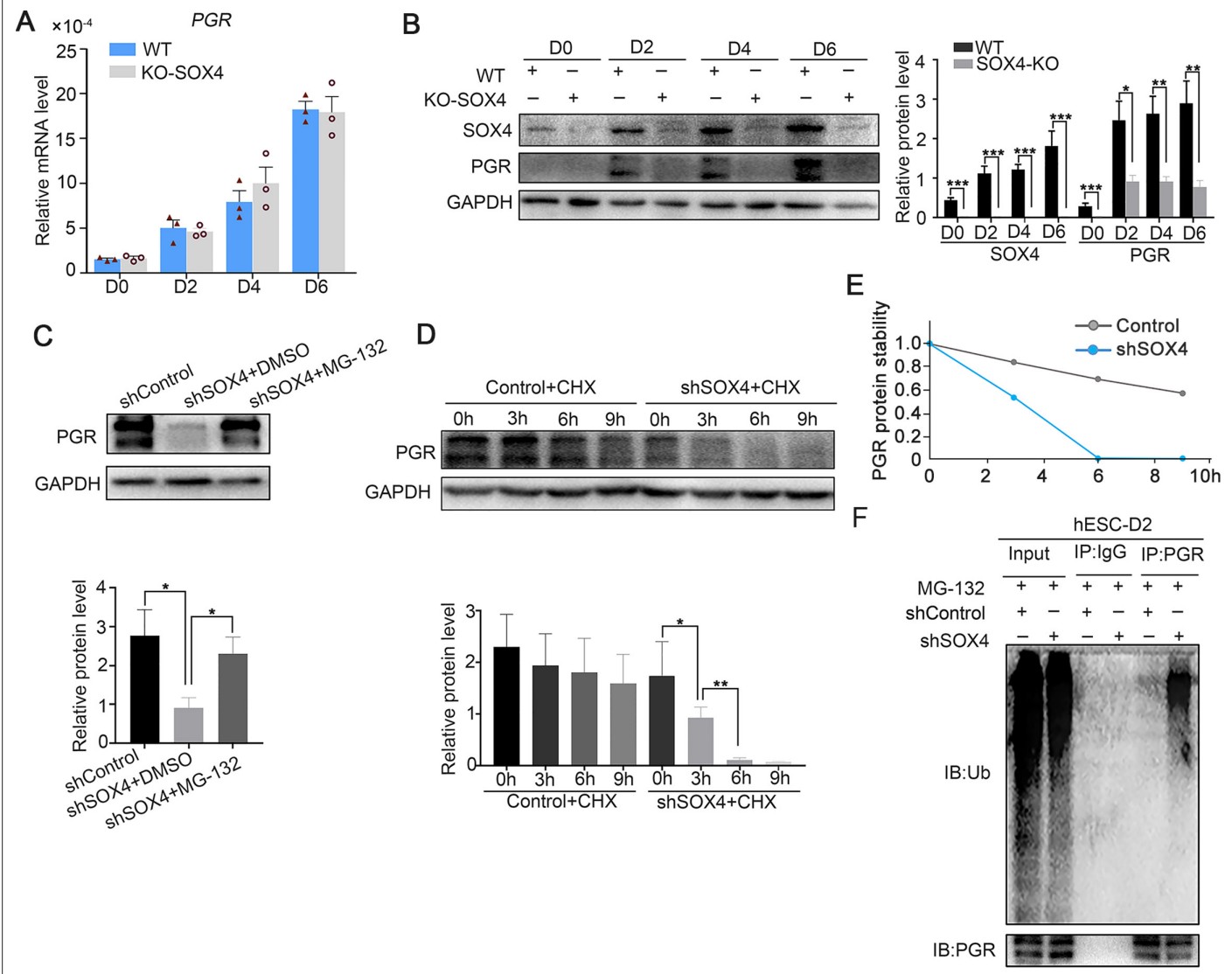

**Figure 5.** SOX4 stabilizes progesterone receptor (PGR) through repressing the ubiquitin–proteasome pathway. (**A**) mRNA levels of *PGR* after SOX4 knockout in decidualized immortalized human endometrial stromal cells (HESCs) at indicated time points after the E2, MPA, and cAMP treatment. Results were presented as means ± standard error of the mean (SEM); *n* = 3. (**B**) Protein levels of PGR and SOX4 after SOX4 knockout in decidualized HESCs at indicated time points after the E2, MPA, and cAMP treatment. (**C**) Protein levels of PGR in the presence of shSOX4 or MG132. MG-132 is added 6 hr before collecting the immortalized cells. (**D, E**) Protein levels of PGR in the presence of protein synthesis inhibitor cycloheximide (CHX) in SOX4-knockdown cells. (**F**) PGR ubiquitination after immunoprecipitation with PGR antibody and blotted with ubiquitin antibody in SOX4-knockdown immortalized HESCs decidualized for 2 days. Cells were treated with MG-132 before cell lysis. The value for relative protein level is band quantification of indicated protein, relative to GAPDH. n=3. *p < 0.05; **p < 0.005; ***p < 0.001.

The online version of this article includes the following figure supplement(s) for figure 5:

**Figure supplement 1.** Regulation of SOX4 on progesterone receptor (PGR) in primary decidualized stromal cells.

Since there were 41 lysine residues in the PGR protein, UbiBrowser (http://ubibrowser.ncpsb.org) was applied to predict the potential domain recognized by E3 ligase HERC4 (***Figure 7—figure supplement 1A***). As the latent recognition domains were located in DBD and LBD domains of PGR, PGR was divided into four fragments (F1, F2, F3, and F4) accordingly (***Figure 7A***). Only F3 (DBD domains of PGR) exhibited protein degradation after cotransfected with HERC4 in 293T cells (***Figure 7B***). Point mutation of all Lysine (K) to arginine (R) in DBD region (***Figure 7C***) showed that K588, K613, K617, and K638 were critical for PGR degradation (***Figure 7D***), which was further sustained by incapable PGR ubiquitination mediated by HERC4 in these mutant forms when compared with WT or K565R in 293T

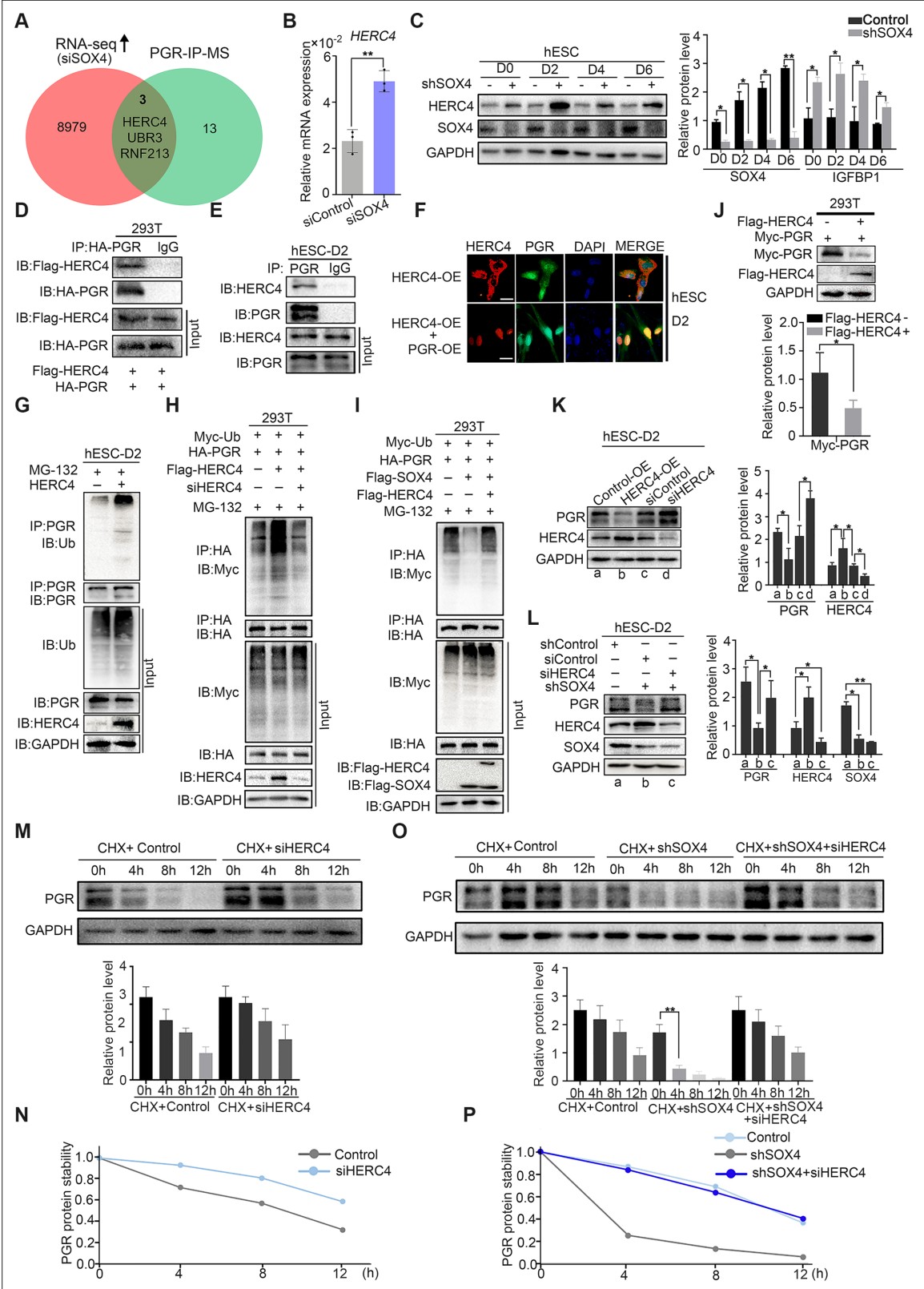

**Figure 6.** SOX4 inhibits progesterone receptor (PGR) degradation by regulating E3 ligase HERC4. (**A**) Venn diagram of overlapping genes between upregulated genes in SOX4 knockdown from RNA sequencing (RNA-Seq) and PGR-associated protein form IP-MS. mRNA and protein levels of HERC4 in the absence of SOX4 by qPCR (**B**) and western blot (**C**). Results were presented as means ± standard error of the mean (SEM); $n = 3$. (**D**) Protein interaction between PGR and HERC4 after overexpression of HA-PGR and Flag-HERC4 in 293T cells. (**E**) Protein interaction between PGR and HERC4

*Figure 6 continued on next page*

*Figure 6 continued*

after overexpression of HA-PGR and Flag-HERC4 in decidualized immortalized human endometrial stromal cells (HESCs) for 2 days, D2: 2 days. (**F**) Localization of PGR and HERC4 after overexpression of PGR and HERC4 in decidualized HESCs for 2 days, scale bar: 100 µm. (**G**) PGR ubiquitination at the present of MG-132 after overexpression of HERC4 in decidualized HESCs for 2 days.(**H**) PGR ubiquitination at the present of MG-132 and ubiquitin with HERC4 overexpression or knockdown in 293T cells. (**I**) PGR ubiquitination at the present of MG-132, ubiquitin, and SOX4 with or without HERC4 in 293T cells. (**J**) Degeneration of PGR after HERC4 overexpression in 293T cells. (**K**) PGR protein levels at the present of HERC4 overexpression and in HERC4 knockdown decidualized immortalized HESCs for 2 days. (**L**) Protein levels of PGR, HERC4, and SOX4 after SOX4 and/or HERC4 knockdown in decidualized immortalized HESCs for 2 days. (**M, N**) The protein half-life of PGR after HERC4 knockdown at the present of protein synthesis inhibitor cycloheximide (CHX) in 2 days decidualized immortalized HESCs. (**O, P**) The protein half-life of PGR after SOX4 and/or HERC4 knockdown at the present of protein synthesis inhibitor CHX in 2 days decidualized immortalized HESCs. The value for relative protein level is band quantification of indicated protein, relative to GAPDH. The above experiments were repeated three times. n=3; *p < 0.05; **p < 0.005.

The online version of this article includes the following figure supplement(s) for figure 6:

**Figure supplement 1.** The SOX4 regulated ubiquitin E3 ligase expression during the endometrial stroma decidualization.

cells (*Figure 7E*). Collective, these results suggested that K588, K613, K617, and K638 were vital for PGR ubiquitination by HERC4.

## Aberrantly decreased endometrial SOX4 expression is associated with recurrent implantation failure undergoing IVF treatment

There was considerable evidence indicating that HESCs decidualization was severely impaired in patients withthe EMS, both in eutopic and ectopic lesions (*Klemmt et al., 2006*). Recurrent implantation failure (RIF) has also been reported to be associated with compromised decidualization (*Zhou et al., 2019*). We next analyzed the expression levels of SOX4, PGR, FOXO1, HERC4, IGFBP1, and PRL in the mid-secretory endometrium of healthy women (control, $n = 12$) versus women who suffered RIF due to endometriosis (EMS-RIF, $n = 12$). The expressions of *SOX4*, *FOXO1*, *IGFBP1*, and *PRL* were significantly decreased in EMS-RIF group with increased HERC4 compared with the control, but the mRNA level of *PGR* was comparable in both groups (*Figure 8A–F*). At protein level, a large portion of endometrial samples from women with EMS-RIF showed reduced expression of SOX4, FOXO1, and IGFBP1. Although the *PGR* mRNA levels were comparable between healthy and RIF samples, PGR protein was significantly decreased accompanied by increased HERC4 in EMS-RIF group (*Figure 8G, H*). Immunostaining analysis further revealed significantly reduced PGR, SOX4, IGFBP1, and FOXO1 expression and increased HERC4 in stromal cells in women with EMS-RIF (*Figure 8I*).

Additionally, primary HESCs were obtained from the proliferative phase endometrium of three healthy and three EMS-RIF patients (*Figure 8—figure supplement 1A*). The expression levels SOX4, FOXO1, PGR, and IGFBP1 were lower in the primary HESCs of EMS-RIF cultured with EPC for 4 days compared to the control (*Figure 8J*). Moreover, overexpression of SOX4 and/or PGR at least partially restored FOXO1 and IGFBP1 expression in EMS-RIF stromal cells (*Figure 8K*). Collectively, our in vivo and in vitro evidence strongly suggested the crucial role of SOX4 in decidualization and female fertility.

## Discussion

Here, we find that SOX4 is the most abundant SOX family member in HESCs and plays a vital role in decidualization. Previous studies have demonstrated that the SOX family controls cell fate and differentiation in various developmental processes (*She and Yang, 2015*; *Angelozzi and Lefebvre, 2019*). A previous study has confirmed that epithelial SOX17, one of SOX transcription factors, is indispensable for uterine receptivity and embryo implantation by regulating IHH expression (*Wang et al., 2018*). The embryo adhesion of implantation process occurred between the embryo and epithelia, and the proper differentiation of epithelia is required for this event (*Ye, 2020*). Moreover, for the endometrial receptivity establishment, the stromal compartment also needs to differentiate in response to the endocrine hormone progesterone and estrogen that is known as decidualization, and the crosstalk between the stroma and epithelia has been widely approved to participate to the establishment of endometrial receptivity (*Li et al., 2011*).

In the present study, we uncover the critical role of SOX4 in stromal cell decidualization by RNA-Seq and ChIP-Seq. Several previously unappreciated genes of SOX4 are unveiled through RNA-Seq.

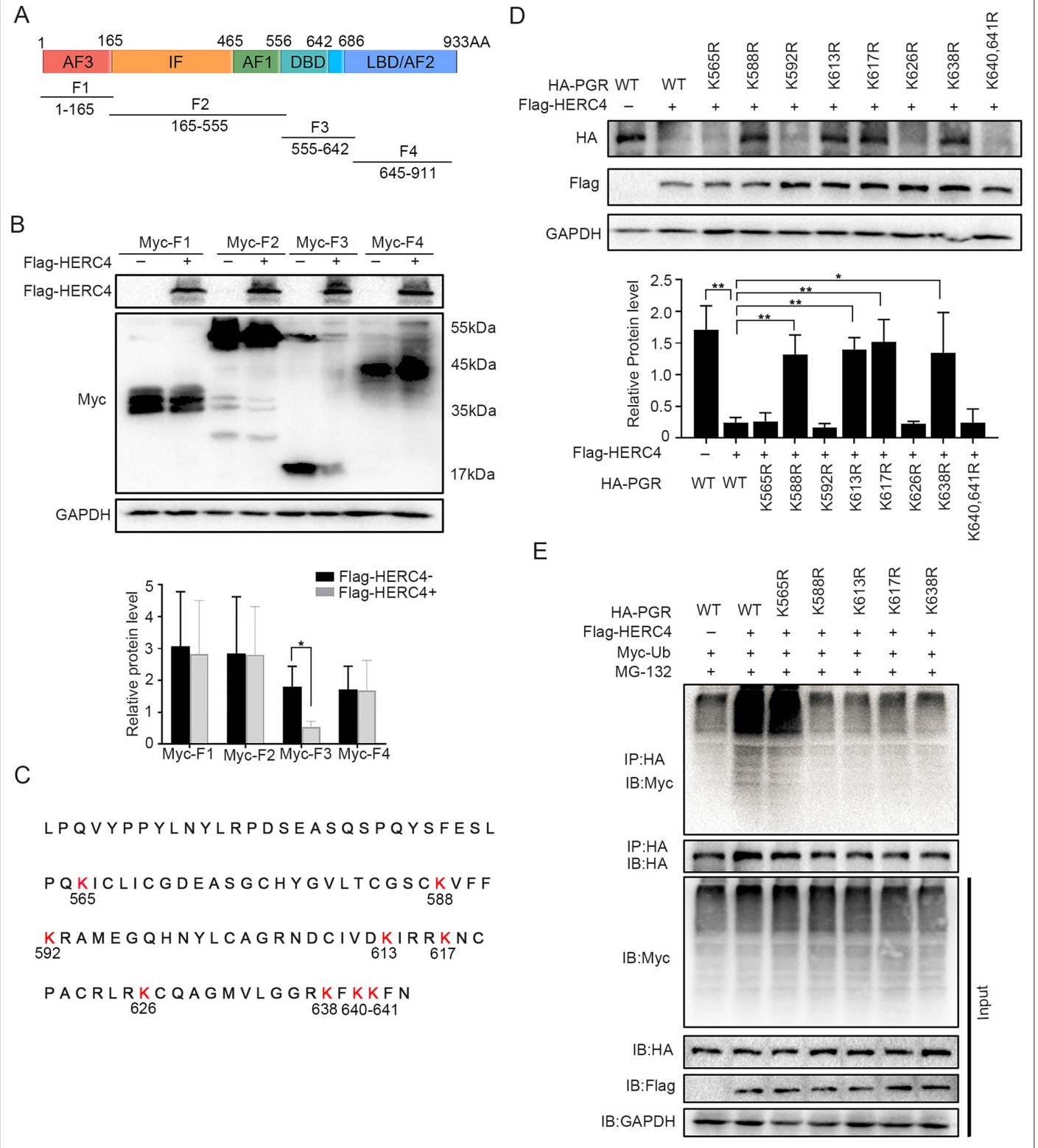

**Figure 7.** HERC4 mediates progesterone receptor (PGR) ubiquitination at K588, K613, K617, and K638. (**A**) Schematic diagram of PGR structure. F1–F4 represents four different functional domains, respectively. (**B**) Protein levels of different fragments of PGR (Myc-tagged F1–F4) at the present of HERC4 in 293T cells. (**C**) Lysine (K) sites in PGR DBD domain. (**D**) Protein levels of PGR after Lysine mutant to Arginine in DBD with HERC4 overexpression in 293T cells.The value for relative protein level is band quantification of indicated protein, relative to GAPDH. (**E**) PGR ubiquitination after Lysine mutant to Arginine in DBD at the present of HERC4, ubiquitin and MG-132 in 293T cells. n=3; *p < 0.05; **p < 0.005.

*Figure 7 continued on next page*

*Figure 7 continued*

The online version of this article includes the following figure supplement(s) for figure 7:

**Figure supplement 1.** The predicted lysine sites for the ubiquitin modification in progesterone receptor (PGR) protein by the E3 ligase HERC4.

FOXO1 is indispensable in the process of decidualization account for the widely overlapping of binding peaks with PGR (*Vasquez et al., 2015*). In this investigation, FOXO1 is regulated by SOX4 at transcriptional level, indicating the essential role of SOX4 in decidualization. We are also very surprised to notice that the direct regulation of SOX4 on FOSL2, an important part of AP1 complex involving inflammation (*Renoux et al., 2020*), expression directly. Moreover, we also find that FOSL2 and FRA2 motifs are significantly enriched in SOX4-binding sites, suggesting the complex interplay between these factors. The regulation of SOX4 on FOLS2 implicates an alternative role of SOX4 regulating decidualization, which deserves further investigation.

PGR is a master regulator of the decidualization process since decidualization is mainly influenced by the progesterone–PGR signaling. After dimerization and nuclear translocation, PGR protein interacts with the following proteins to synergistically direct the differentiation program: FOXO1, C/EBPβ, STAT3, STAT5, HOXA10, and HOXA11 (*Gellersen and Brosens, 2014*). An array of modulators regulates the functional plasticity of PGR, including subcellular distribution, protein modification, and interaction with coregulators (*Wu et al., 2018*). In this study, we were intrigued by the finding that SOX4 affects PGR protein stability rather than its transcription.

During the menstrual cycle, estrogen–ER signaling induced endometrial PGR mRNA in the proliferative phase, recapitulating the regulation manner of PGR in mouse uteri (*Tan et al., 1999*; *McKinnon et al., 2018*). Here, we provided evidence that PGR protein is further stabilized by increased SOX4 at post-transcriptional regulation in the later secretory phase. Previous studies have validated that SOX4 regulates P53 stability through E3 ligase Mdm2 (*Pan et al., 2009*). In breast cancer cells, E3 ligase BRCA1 (*Calvo and Beato, 2011*) and CUEDC2 (*Zhang et al., 2007*) have been shown to regulate PGR protein stability. Multiple evidence in our research has verified that the E3 ligase HERC4 functions as a ubiquitin-modified enzyme for the PGR protein. Further exploration revealed that lysine residual in the PGR DNA-binding domain possess modified sites for ubiquitination. In the endometrial cells, there are very limited reports regarding PGR protein modification. Our previous study proved that Bmi1 facilitates the PGR ubiquitination through E3 ligase E6AP, which promotes PGR transcriptional activity instead of protein degradation (*Xin et al., 2018*). Besides, SUMOlation of PGR has been reported to occur at the K388 site and fine-tunes the transcriptional activity of PGR (*Jones et al., 2006*). The Lys-388 is a key site not only for PGR-B SUMOlation, but also a key site for CUEDC2-mediated ubiquitination in proteasome in breast cancer (*Zhang et al., 2007*). Here, we characterized that K588, K613, K617, and K638 were critical sites for E3 ligase HERC4, which mediated PGR ubiquitination and degradation. There were other E3 ligases interacting with PGR as revealed by our mass spectrum, whether these E3 ligases also mediated PGR ubiquitination remains largely unknown. Multiple specific E3 ligases may exist for PGR ubiquitination rely on individual lysine residues.

Here, we report a previously unrecognized PGR ubiquitination and degradation modified by the ubiquitin E3 ligase HERC4 regulated by SOX4 in endometrial stromal cells. There are several potential postulations concerning the underlying mechanism by which SOX4 inhibits HERC4 expression in stromal cells. SOX4 has been reported to interact with repressive histone modifiers, such as H3K27 trimethylation enzyme EZH2, to directly repress the target gene transcription (*Koumangoye et al., 2015*). On the other hand, SOX4 indirectly represses the target gene expression through regulating EZH2 expression (*Tiwari et al., 2013*). Since no direct SOX4 binding on HERC4 is observed, the detailed mechanism by which SOX4 regulates HERC4 transcription in endometrial stroma cells requires further exploration.

Considering the critical role of progesterone during the pregnancy establishment and maintenances, both natural and synthetic progestogens have been widely used to improve endometrial function in women with a history of RIF and unexplained recurrent pregnancy loss. However, some of these patients still suffered from RIF and recurrent miscarriages due to progesterone resistance. The causes of progesterone resistance may be related to defects in the PGR signaling and its molecular chaperones as well as decreasing PGR transcriptional activity (*McKinnon et al., 2018*). PGR functional deficiency is related to abnormal PGR mRNA levels, post-transcriptional modifications, post-translational

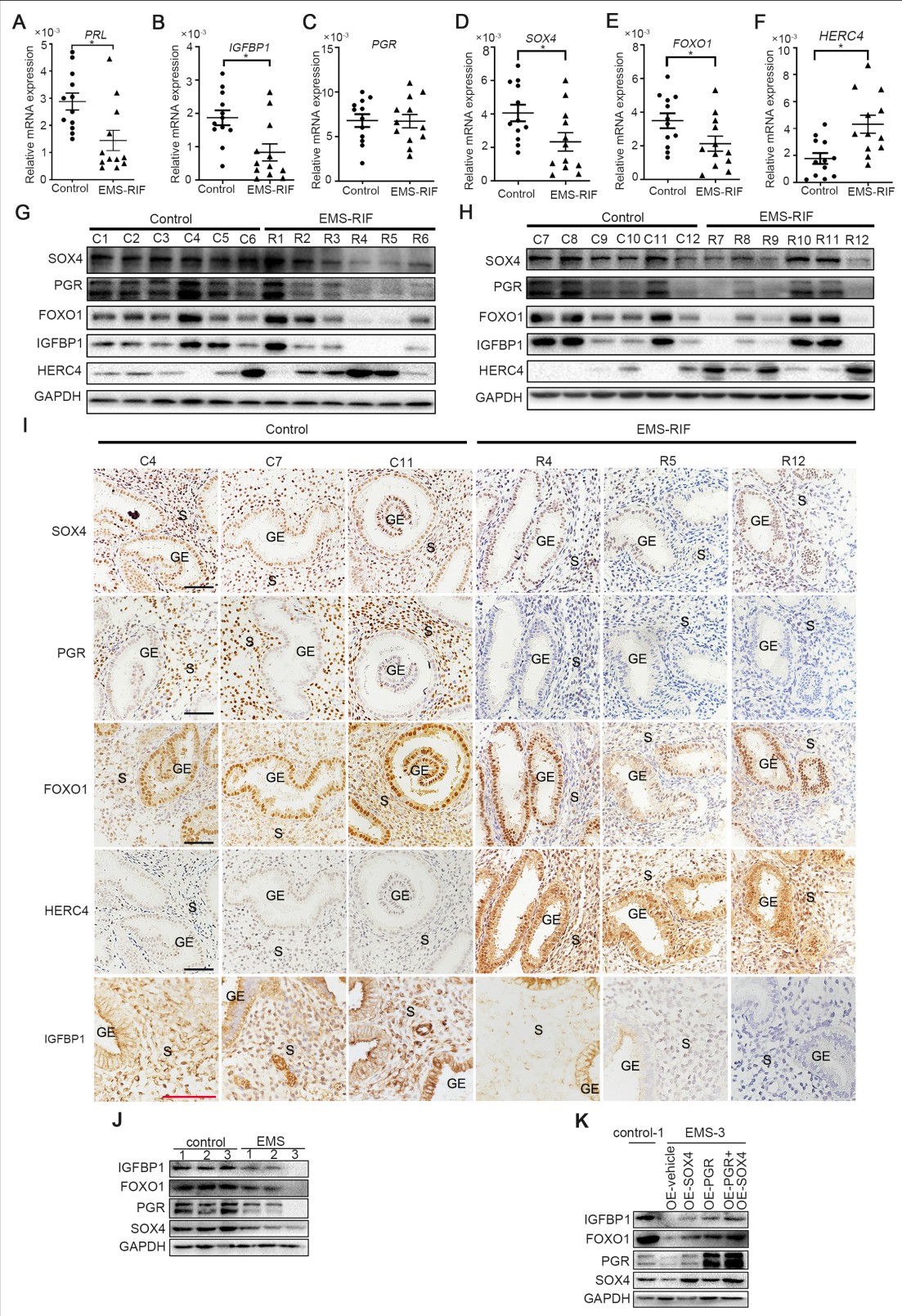

**Figure 8.** Endometrial SOX4 expression in RIF of women with EMS undergoing IVF treatment. (**A–F**) Expression of *SOX4*, *FOXO1*, *IGFBP1*, *PRL*, *PGR*, and *HERC4* in mid-secretory endometrium from control (*n* = 12) and EMS-RIF (*n* = 12). Results were presented as means ± standard error of the mean (SEM). *p < 0.05. (**G, H**) Protein levels of SOX4, FOXO1, IGFBP1, PGR, and HERC4 in mid-secretory endometrium from control (*n* = 12) and EMS-RIF (*n* = 12). C1–C12 and R1–R12 represent tissues from different patient. (**I**) Localization of SOX4, FOXO1, IGFBP1, PGR, and HERC4 in control (*n* = 3) and EMS-

*Figure 8 continued on next page*

*Figure 8 continued*

RIF groups as detected by immunostaining. C4, C7, C11, R4, R5, and R12 represent tissues from different patient. GE: gland epithelium; S: stroma. Scale bar: 100 μm. (**J**) Protein levels of SOX4, IGFBP1, FOXO1, and PGR in primary human endometrial stromal cells (HESCs) of control and EMS by western blot. (**K**) Protein levels of IGFBP1, FOXO1, PGR, and SOX4 in decidualized primary endometrial stromal cells from EMS-3 after overexpression of SOX4 and/or PGR.

The online version of this article includes the following figure supplement(s) for figure 8:

**Figure supplement 1.** Histology of pelvic endometriosis tissue.

modifications, and protein stability (*Xin et al., 2016*; *McKinnon et al., 2018*). However, the specific causes of PGR defects are, at present, poorly explored and understood.

EMS is frequently accompanied by progesterone resistance and infertility. Several transcription factors, including the factors in GATA and HOX family can regulate the progesterone signaling, and the epigenetic differential methylation of PR-B, HOX, and GATA family transcription factor can lead to their decreased expression, leading to the disturbed progesterone signaling in the EMS (*Longo et al., 2020*). It is interesting that lower SOX4 expression is strongly relevant with reduced PGR protein expression in endometrial stromal cells of EMS who experience RIF, and the level of PGR is restored to some extent when SOX4 is overexpressed in these stromal cells. It is conceivable that PGR defects caused by insufficient SOX4 may potentially result in implantation failures, even with high doses of progestogens supplementation during IVF. In our previous study using human endometrial samples from different patients with recurrent spontaneous abortion cohort, we also observed progesterone resistance as revealed by defective PGR signaling, with normal PGR protein level but diminished transcriptional PGR activity (*Xin et al., 2018*). This implies that any disturbances in the transduction of P4-PGR signaling pathway will severely influence the endometrial cell responsiveness to progesterone and contribute to infertility.

Collectively, our investigation provides compelling evidence that SOX4 plays a key role in HESCs decidualization through the transcriptional regulation of critical factors related to decidualization. Meanwhile, SOX4 endows human stromal cells appropriate progesterone responsiveness by fine-tuning PGR protein stability. Aberrant SOX4 expression is strongly associated with decreased PGR and FOXO1 expression in the endometrium of women who have experienced RIF with EMS, implying the high clinic relevance of SOX4 in female fertility and pregnancy maintenance. A better understanding of the regulatory network of SOX4 will facilitate the development of therapeutic strategy for the clinical treatment of RIF in EMS.

## Materials and methods

### Key resources table

| Reagent type (species) or resource | Designation | Source or reference | Identifiers | Additional information |
|---|---|---|---|---|
| Cell line (*H. sapiens*) | T HESCs | ATCC | CRL-4003 RRID: CVCL_C464 | |
| Antibody | Anti-SOX4 (Rabbit polyclonal) | Abcam | Cat # ab80261 RRID:AB_1658989 | WB (1:600) ICC (1:1000) IF (1:300) |
| Antibody | Anti-SOX4 (Rabbit polyclonal) | Diagenode | Cat # C15310129 | IP (1:100) |
| Antibody | Anti-PGR (Rabbit monoclonal) | CST | Cat # 8757 RRID:AB_2797144 | WB (1:500) ICC (1:3000) IF (1:200) IP (1:50) ChIP (1:50) |

*Continued on next page*

*Continued*

| Reagent type (species) or resource | Designation | Source or reference | Identifiers | Additional information |
|---|---|---|---|---|
| Antibody | Anti-HERC4 (Rabbit polyclonal) | Protein tech | Cat # 13691-1-AP RRID:AB_10596480 | WB (1:1000) ICC (1:1000) IF (1:300) |
| Antibody | Anti-FOXO1 (Rabbit polyclonal) | Abcam | Cat # ab39670 RRID:AB_732421 | WB (1:1000) ICC (1:3000) |
| Antibody | Anti-IGFBP1 (Rabbit polyclonal) | Abcam | Cat # ab228741 | WB (1:1000) IHC (1:500) |
| Antibody | Anti-Ub (Mouse monoclonal) | CST | Cat # 3936 RRID:AB_331292 | WB (1:1000) |
| Antibody | Anti-HA (Rabbit monoclonal) | CST | Cat # C29F4 | WB (1:1000) IP (1:100) ChIP (1:100) |
| Antibody | Anti-Myc (Mouse monoclonal) | CST | Cat # 2276 RRID:AB_331783 | WB (1:1000) |
| Antibody | Anti-Flag (Mouse monoclonal) | Sigma | Cat # F9291 RRID:AB_439698 | WB (1:1000) |
| Antibody | Anti-GAPDH (Rabbit polyclonal) | Abmart | Cat # P30008 | WB (1:2000) |
| Chemical compound, drug | E2 | Sigma | E8875 | Final concentration: 10 nM |
| Chemical compound, drug | MPA | Sigma | M1629 | Final concentration: 1 μM |
| Chemical compound, drug | cAMP | MCE | HY-B0764 | Final concentration: 0.5 mM |
| Chemical compound, drug | MG132 | Selleck | S2619 | Final concentration: 20 μM |
| Chemical compound, drug | 3-MA | Selleck | S2767 | Final concentration: 20 mM |
| Chemical compound, drug | CHX | MedChemExpress | HY-12320 | Final concentration: 20 μg/ml |
| Chemical compound, drug | Insulin–transferrin–selenium | Thermo Fisher | 51500-056 | Final concentration: 1% |
| Chemical compound, drug | Puromycin | MedChemExpress | HY-15695 | Final concentration: 500 ng/ml |
| Chemical compound, drug | RNAi MAX | Thermo Scientific | 13778030 | 7.5 μl/6-well |
| Chemical compound, drug | Lipofectamine 2000 | Thermo Scientific | 11668030 | 5 μl/6-well/2500 ng DNA |
| Commercial assay or kit | Dual-Luciferase Reporter Assay System | Promega | E1910 | |
| Commercial assay or kit | ChIP-IT high Sensitivity | Active Motif | 53,040 | |
| Commercial assay or kit | KAPA DNA HyperPrep Kit | Roche | KK8502 | |

## Sample of clinical cases

Healthy women who had already gave birth and infertile women with RIF due to EMS were recruited from Liuzhou Maternity and Child Health Care Hospital in China from March 2018 to December 2020. The study had been approved by hospital ethics committee and all participants signed informed consent. The age of participates is between 20 and 38 years of age with body mass index between 18 and 23. The thickness of the endometrium on ovulation days was between 8 and 16 mm with menstrual cycle between 28 ± 7 days and no steroid treatment or other medication for at least 2–3 months before biopsy. Detailed information of patients is listed in *Supplementary file 1*. Patients with polycystic ovary syndrome, endometrial polyps, chronic endometritis, and hydrosalpinges were excluded. Only endometrial tissue with no apparent pathology assessed by a pathologist was kept for further experiments. Decidual tissue in early pregnancy was collected from the women who underwent legal termination of an apparently normal early pregnancy (about 8 weeks). RIF referred to the inability to achieve a clinical pregnancy after transferring at least four good-quality embryos in a minimum of three fresh or frozen cycles in a woman under 40 years old (*Coughlan et al., 2014*).

## Isolation and culturing of primary endometrial stromal cells

Primary HESCs were obtained from healthy, reproductive-aged volunteers with regular menstrual cycles or EMS stage IV patients with infertility. An endometrial biopsy was performed during the proliferative phase of the menstrual cycle. These participants were recruited from The First Affiliated Hospital of Xiamen University in China from July 2019 to October 2020. Staging of EMS was according to rAFS (*The American Fertility Society, 1985*). Detailed information of patients is listed in *Supplementary file 1*, and pathology of pelvic EMS is shown in *Figure 8—figure supplement 1A*. The study had been approved by hospital ethics committee and all participants signed informed consent. Participants were documented not under hormone treatments for at least 3 months before surgery. Only endometrial tissue with no apparent pathology assessed by a pathologist was kept for further experiments. Primary HESCs of EMS women were isolated by laparoscopy, and the isolation and culturing of primary endometrial stromal cells were performed as follows. The endometrial tissues were first cut into pieces as small as possible and subjected to type IV collagenase (2% concentration) digestion for 1 hr. Two hours after cell seeding, culture medium was changed to remove the floating cells.

## Establishment of SOX4 knockout cell lines

Immortalized HESCs cell lines were purchased from ATCC Corporation (American Type Culture Collection, ATCC crl-4003). SOX4 knockout was generated by CRISPR/Cas9 approach as previously described (*Zhang et al., 2019*). (1) sgRNAs targeting SOX4 gene (NM_003107.3) were designed on the website (https://zlab.bio/guide-design-resources) and subcloned into the pL-CRISPR.EFS.GFP vector (Addgene plasmid #57818). (2) Cas9 plasmid was packaged with lentivirus. After infecting stromal cells, Cas9 positive cells were sorted and were plated into a 96-well plate with one cell per well. (3) Knockout efficiency was verified by DNA sequencing and western blot after single-cell-derived clone expansion.

## In vitro culture of HESCs andd decidualization

Immortalized HESCs were maintained in DMEM/F12 without phenolic red (Gibco) in the presence of 10% charcoal stripped fetal bovine serum (CS-FBS, Biological Industries), glucose (3.1 g/l), sodium pyruvate (1 mM, Sigma); sodium bicarbonate (1.5 g/l, Sigma), penicillin–streptomycin (50 mg/ml, Solarbio); insulin–transferrin–selenium (1%, Thermo Fisher) and puromycin (500 ng/ml). The primary cultured cells were maintained in DMEM/F12 (Gibco) without phenolic red and 10% CS-FBS. For decidualization, HESCs were cultured in medium at the present of estrogen (E2, 10 nM, Sigma), medroxy-progesterone acetate (MPA, 1 mM, Sigma), and dibutyl cyclophospsinoside (db-cAMP, 0.5 mM, MCE) in 2% CS-FBS with different days. All the cells were cultured in 5% $CO_2$, 95% air, 100% humidity at 37°C and culture medium was replaced every 48 hr. The endometrial stromal cell line was purchased from ATCC and authenticated using STR profiling by ATCC, and tested to be free from mycoplasma contamination.

## Plasmid construction and siRNA transfection

The overexpressed plasmids of SOX4 (pLVML-FLAG-SOX4-IRES-puro and pLVX-HA-IRES-ZSgrenn-SOX4), HERC4 (pLVX-FLAG-HERC4-IRES-Zsgreen, pENTER-FLAG-HERC4, and HA-PCMV-HERC4), PGR (pLVX-MYC-PGR-IRES-Zsgreen and HA-pCMV-PGR), and ubiquitin (pRK5-HA-Ubiquitin-WT and MYC-Ubiquitin-WT) were purchased from Wu Han Miao Ling Company or homemade. The list for plasmid used in this study is shown in *Supplementary file 3*. HESCs were transfected with these plasmids by Lipofectamine 2000 transfection reagent (Invitrogen) or infected with the lentivirus, and 48 hr later, HESCs were decidualized with EPC treatment. ShSOX4 lentivirus was purchased from Shanghai Ji Man Company. The ShSOX4 lentivirus and the control lentivirus were placed in the HESCs medium, 48 hr after infection, HESCs were decidualized with EPC.siRNAs targeting SOX4, HERC4, PGR, and FOXO1 were purchased from Guangzhou RiboBio Biological Company (see the specific sequence for details in *Supplementary file 2*). RNA interference was carried out according to the manufacturer's instructions. Briefly, 10 mM siRNA was transfected into HESC with Lipofectamine RNAiMAX (Invitrogen, Carlsbad, USA). For the knockdown and overexpression experiment, the silencing and overexpression were first performed without differentiation stimulus, and 24 hr later hESCs were decidualization for indicated time.

## ChIP-Seq and ChIP-qPCR

ChIP-Seq was performed according to the manual of ChIP-IT high Sensitivity (Active motif, catalog no. 53040). Briefly, both immortalized HESCs and primary cultured endometrial stroma cells (treated with EPC to induced decidualization for 2 days) with exogenous expressed HA-SOX4 were crosslinked and immunoprecipitated withHA (CST, c29F4) and PGR (CST, 8757) antibodies. Immunoprecipitated and input DNA were quantified using Qubit 4.0 fluorometer. Libraries were prepared using the KAPA DNA HyperPrep Kit (KK8502) and sequenced with an Illumina Nova-PE150. ChIP-qPCR was performed as ChIP-Seq with antibodies HA (CST, c29F4) and PGR (CST, 8757) and the immunoprecipitated DNA was detected by QPCR. All the PCR primers used in ChIP-qPCR are listed in *Supplementary file 2*.

## Immunoprecipitation

For protein interaction, IP experiments were performed as previously described (*Xin et al., 2018*). Anti-PGR (CST, 8757), Anti-HA (CST, c29F4), SOX4 (Diagenode, C15310129), and mouse IgG or rabbit IgG Mouse Anti-Rabbit IgG (Conformation Specific, L27A9 mAb) were used for IP. The immunoprecipitants were washed four times in lysis buffer, resolved with sodium dodecyl sulfate–polyacrylamide gel electrophoresis (SDS–PAGE), and immunoblotted with corresponding antibodies.

## RNA-Seq

Scramble or siSOX4 RNA were transfected into HESCs at 50% confluence with RNAi MAX (Invitrogen). Immortalized HESCs were decidualized for 2 days after transfection and RNAs were collected for RNA-Seq of BGISEQ (China, BGI). Purified RNA was prepared and subjected to 50 bp single-end RNA-Seq.

## IP MS

Immortalized HESCs were used to prepare IP samples after treated with EPC 2 days. After Co-IP with PGR-conjugated agarose beads, the immunoprecipitants were resolved with SDS–PAGE and visualized using Coomassie brilliant blue stain. The discrete bands between PGR and IgG control were isolated, digested, purified, and subjected to liquid chromatograph (LC)–MS in School of Life Sciences, Xiamen University.

## Dual-luciferase reporter assay

Construction of luciferase reporter was performed as previously described (*Jiang et al., 2015*). The promoter regions of SOX4 were amplified from genomic DNA and subcloned into pGL3 plasmid. All constructs were transiently transfected into 293T cells using Lipofectamine 2000. Total cell lysates were prepared 36 hr after transfection, and luciferase activity was measured using the Dual-Luciferase Reporter Assay System (Promega Corporation). Firefly luciferase activity was normalized by Renilla luciferase activity.

## PGR ubiquitination assay

The ubiquitination assays were performed as previously described (*Xin et al., 2018*). MG-132 (Sigma M8699, 20 µM) was added to the cultured medium 6 hr before cells collection. Cell lysis was immunoprecipitated with antibodies against PGR and HA, respectively. Proteins were released from the beads by boiling in SDS–PAGE sample buffer, and ubiquitination was analyzed by immunoblotting with different antibodies.

## Protein stability experiment and degradation assay

To determine the protein stability of PGR protein, SOX4-knockdown and control immortalized HESCs decidualized for 2 days followed by treatment with CHX (MedChemExpress, 20 µg/ml), a protein synthesis inhibitor, for 0, 3, 6, 9 hr or 0, 4, 8, 12 hr. The protein degradation assay of PGR, the immortalized HESCs with 50% confluency were induced decidualization for 2 days, followed by treatment with the proteasome inhibitor MG132 (Selleck, 20 µM, 6 hr), and the autophagy inhibitor 3-MA Selleck, 20 mM, 6 hr. A group treated with Dimethylsulfoxide (DMSO) served as the negative control group.

## Immunostaining

For healthy endometrial tissue, proliferative phase samples were timed based on the patient's cycle day, and luteal phase samples were timed using the subject's urinary luteinizing hormone surge.

All the samples from the biopsy were fixed in formalin and embedded in paraffin for section. After deparaffinization and hydration, formalin-fixed paraffin embedded endometrial sections (5 µm) were subjected to antigen retrieval by autoclaving in 10 mM sodium citrate solution (pH = 6.0) for 10 min. A diaminobenzidine (Sigma) solution was used to visualize antigens. Sections were counterstained with hematoxylin. In immunofluorescence studies, formalin-fixed HESCs cells were blocked with 5% Bovine Serum Albumin (BSA) in PBS and immune stained by antibodies for SOX4 (Abcam,ab80261), PGR (CST, 8757), FOXO1 (Abcam, ab39670), IGFBP1 (Abcam, ab228741) and HERC4 (Proteintech, 13691-1-AP). Signals were visualized by secondary antibody conjugated with Cy2 or Cy3 fluorophore (Jackson Immunoresearch). Sections were counterstained with Hoechst 33,342 (2 µg/ml, Life Technology).

## Real-time PCR

Quantitative real-time PCR was performed as described previously (*Jiang et al., 2015*). Total RNA was extracted from HESCs or endometrium using TRIzol reagent (Invitrogen) following the manufacturer's protocol. The quality of RNA was determined by electrophoresis and concentration of RNA was measured by Nanodrop. A total of 1 µg RNA was used to synthesize cDNA. Quantitative real-time PCR was performed with SYBR Green (Takara) on an ABIQ5 system. All expression values were normalized against GAPDH. All PCR primers are listed in *Supplementary file 2*.

## Western blot analysis

Western blot analysis was performed as described previously (*Zhang et al., 2019*). Antibodies against SOX4, IGFBP1, FOXO1, PGR, HERC4, FOSL2, HA, Myc, Flag, and ubiquitin were used. GAPDH served as a loading control. The value for relative protein level represents the quantification of band intensity of indicated protein with the loading control GAPDH using Image J.

## Bioinformatic analysis

The softwares for RNA-Seq and ChIP-Seq analysis including STAR (2.7.3a) for alignment, MACS2 (2.2.7.1) for peakcall of ChIP-Seq, ggplot2 (3.3.5) for visualization, ngs.plot.r (2.61) for peak heatmap of ChIP-Seq and the packages of edgeR (3.9), Complex heatmap (2.4.3), ChIPseeker (1.24.0) in R (4.1). For ChIP-Seq, after filter the raw data to remove adapter from read by Trimgalore, the clean reads were aligned to the human genome (Hg38) by HISAT2. Only uniquely aligned reads were kept for downstream analysis. MACS2 was applied for peak call using default parameters. The enrichment of SOX4 binding on specific gene was visualized in Integrative Genomics Viewer (IGV). RNA-Seq raw data were initially filtered to obtain clean data after quality control by Trimgalore. Clean data were aligned to the human genome (Hg38) by HISAT2. RPKM value of each gene was calculated by EdgeR package in R.

## Statistics

Statistical analysis was performed with GraphPad Prism. The data were shown as the mean ± standard error of the mean. Statistical analyses were performed using two-tailed Student's *t*-test or analysis of variance. All experiments were repeat at least three times and p values less than 0.05 were considered statistically significant.

## Acknowledgements

This work was supported by National Key R&D program of China (2021YFC2700302 to H.W., 2018YFC1004404 to S.K.), National Natural Science Foundation of China (81830045 and 82030040 to H.W., 81960280 to A.Q., 82001553 to P.H., 81701483 and 81971419 to W.D.), Fundamental Research Funds for the Central Universities (20720190073 to W.D.), and Guangxi Natural Science Foundation Project (2019JJB140179 to P.H.). Thanks to Zhixiong Huang and Qionghua Chen for their helpful assistant during collecting the endometrial sample from patients with endometriosis.

# Additional information

## Funding

| Funder | Grant reference number | Author |
| --- | --- | --- |
| National Key R&D program of China | 2021YFC2700302 | Haibin Wang |
| National Key R&D program of China | 2018YFC1004404 | Shuangbo Kong |
| National Natural Science Foundation of China | 81830045 | Haibin Wang |
| National Natural Science Foundation of China | 82030040 | Haibin Wang |
| National Natural Science Foundation of China | 81960280 | Aiping Qin |
| National Natural Science Foundation of China | 82001553 | Pinxiu Huang |
| National Natural Science Foundation of China | 81701483 | Wenbo Deng |
| National Natural Science Foundation of China | 81971419 | Wenbo Deng |
| Fundamental Research Funds of the Central Universities | 20720190073 | Wenbo Deng |
| Guangxi Natural Science Foundation project | 2019JJB140179 | Pinxiu Huang |

The funders had no role in study design, data collection, and interpretation, or the decision to submit the work for publication.

## Author contributions

Pinxiu Huang, Data curation, Formal analysis, Investigation, Methodology, Writing – original draft; Wenbo Deng, Formal analysis, Funding acquisition, Resources, Software, Writing - review and editing; Haili Bao, Mengying Liu, Xiaobo Zhou, Manting Qiao, Investigation; Zhong Lin, Yihua Yang, Jingsi Chen, Dunjin Chen, Resources; Jinxiang Wu, Han Cai, Methodology; Faiza Rao, Writing - review and editing; Jinhua Lu, Data curation, Formal analysis; Haibin Wang, Shuangbo Kong, Conceptualization, Formal analysis, Funding acquisition, Supervision, Writing - review and editing; Aiping Qin, Conceptualization, Funding acquisition, Project administration, Resources

## Author ORCIDs

Mengying Liu http://orcid.org/0000-0002-5987-1287
Haibin Wang http://orcid.org/0000-0002-9865-324X
Shuangbo Kong http://orcid.org/0000-0002-7513-4041

## Ethics

The studies for collecting the endometrial tissue were approved by The Ethical Committee of the Faculty of the Liuzhou Maternity and Child Health Hospital (2020-074). The studies for isolating the primary endometrial cells were approved by The Ethical Committee of The First Affiliated Hospital of Xiamen University (XMYY-2021KYSB044). All participants signed informed consent.

## Decision letter and Author response

Decision letter https://doi.org/10.7554/eLife.72073.sa1
Author response https://doi.org/10.7554/eLife.72073.sa2

# Additional files

## Supplementary files
- Supplementary file 1. Clinical information of all patients and controls.
- Supplementary file 2. All primers used in this study.
- Supplementary file 3. The plasmids used in this study.
- Transparent reporting form

## Data availability
The sequencing data generated in this study have been deposited in the Gene Expression Omnibus database under accession code GSE146280 (RNA-Seq), GSE174602 (ChIP-Seq in stromal cell line) and GSE174602 (ChIP-Seq in primary cultured stromal cell).

The following datasets were generated:

| Author(s) | Year | Dataset title | Dataset URL | Database and Identifier |
|---|---|---|---|---|
| Deng W | 2020 | Gene expression in human endometrium cells (ATCC 4003) after Sox4 was abolished by siRNA | https://www.ncbi.nlm.nih.gov/search/all/?term=GSE146280 | NCBI Gene Expression Omnibus, GSE146280 |
| Deng W | 2022 | Gene expression in human endometrium cells (ATCC 4003) after Sox4 was abolished by siRNA (ChIP-Seq in stromal cell line) | https://www.ncbi.nlm.nih.gov/search/all/?term=GSE174602 | NCBI Gene Expression Omnibus, GSE174602 |

The following previously published dataset was used:

| Author(s) | Year | Dataset title | Dataset URL | Database and Identifier |
|---|---|---|---|---|
| Deng W, Dey S | 2019 | Gene expression in human endometrium cells (ATCC 4003) after Sox4 was abolished by siRNA (ChIP-Seq in primary cultured stromal cell). | https://www.ncbi.nlm.nih.gov/search/all/?term=GSE116096 | NCBI Gene Expression Omnibus, GSE116096 |

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
