## [Editor Report]

The manuscript provides a novel mechanism of progesterone receptor stability mediated by SOX4 in human endometrial decidualization. The authors have addressed all the concerns raised by the reviewers and in addition provided the gels at better resolution, leading to a significantly improved paper.

---

## [Decision Letter]

**Decision letter after peer review:**

Thank you for submitting your article "SOX4 facilitates PGR protein stability and FOXO1 expression conducive for human endometrial decidualization" for consideration by *eLife*. Your article has been reviewed by 3 peer reviewers, and the evaluation has been overseen by a Reviewing Editor and Mone Zaidi as the Senior Editor. The reviewers have opted to remain anonymous.

Your manuscript has been evaluated by 3 independent experts. All of them have agreed that your manuscript is very interesting and provides new knowledge to the field. They also raised several concerns including the use of immortalized cells for ChipSeq studies instead of native stromal cells, lack of methodological details for some of the experiments, lack of full-description of differentially regulated genes, incomplete clinical data and a clear rationale for the reported studies.

Based on your manuscript, the reviews and your responses, we invite you to submit a revised version incorporating the revisions as outlined in your response to the reviews.

Some Essential Revisions that require your careful attention are listed below:

– Expand Materials and methods section to provide more experimental details.

– Provide Rationale for Sox4 ChIP-seq using immortalized cells and provide details of timing for ChIP-seq studies.

– Provide data on knock down of SOX4 and HERC4 to rescue decidualization gene expression.

– Indicate unregulated genes in the absence of SOX4 in ChIP-seq experiments.

– Clarify discussion on receptivity and decidualization.

– Provide more clinical data including sample size, characteristics and sample collection periods.

– Numerous methodological details need to be provided as suggested by Reviewer 3.

Additional detailed comments by the Reviewers are listed below. Please address them point-by-point in your Revised manuscript.

*Reviewer #1 (Recommendations for the authors):*

(1) An overarching issue is that the Results and Figure legends are not entirely descriptive about the experiments conducted to generate the data provided in the figures. The reader must delve into the Materials and methods to figure out how the experiment was conducted in terms of the approach. This issue can be rectified by providing more detail on the experimental approach for each major investigation and result. The Materials and methods need to be very clear on how experiments were conducted and analyzed.

(2) Line 188: What is the rationale for not using a SOX4 antibody to conduct the ChIP-seq studies in non-immortalized cells as opposed to overexpression of using a transfection approach? More detail needs to be presented on the treatments and timing of SOX4 ChIP-seq.

(3) Figures 6M and 6N: The figure legend indicates that this experiment was repeated three times, but the graphs do not provide error bars. Was the data for each Western blot quantified with respect to PGR protein and normalized to GAPDH? If so, why are error bars not present on the graphs?

(4) Supplemental Figure 4: The authors contend that HERC4 overexpression decreases PGR abundance, but the level of PGR protein is not assessed, and the PGR immunolocalization data is not very convincing.

(5) Functional studies should be conducted to rescue decidualization gene expression with knockdown of HERC4 and SOX4. Alternatively, the authors can express a HERC4 resistant form of PGR and investigate the impact of SOX4 on decidualization.

(6) Figure 8: The data on EMS-3 are very intriguing but incomplete. If IGFBP1 is enhanced by SOX4 and PGR overexpression, then FOXO1 should also be investigated as well as PRL and other SOX4 and/or PGR responsive genes. In essence, the endometriosis angle for this patient is incomplete as presented.

(7) The manuscript would benefit from editing to correct problems with English grammar and usage.

(8) Line 356: HOX010 and HOXO11 should be HOXA and not HOXO.

(9) Discussion: The fact that different patients with endometriosis-associated RIF have varying defects in several transcription factors should be highlighted in the Discussion. In essence, the underlying variability in several of the key transcription factors can all lead to the syndrome of progesterone resistance.

*Reviewer #2 (Recommendations for the authors):*

1) In Figure 1, the authors have examined the SOX4 expression during the menstrual cycle, how about its expression in the decidual tissue during the pregnancy?

2) In Figure 4 The author showed the overlap the down-regulated genes in the absence of SOX with the ChIP data, how about the up-regulated genes?

3) In Figure 4 the motif analysis for SOX4 ChIP-seq suggested the potential co-binding of regulatory region between the SOX4 and other proteins such as FOSL2, did the author tried to test the protein physical interaction during the decidualization?

4) In Figure 5, while the proteasome inhibitor MG-132 could almost rescue the PR protein degradation in the SOX4 shRNA treatment, did the authors consider whether other protein degradation system, such as the lysosome mediated protein degradation may also be involved? This may deserve some discussions.

Other suggestions:

1. The "Eichment" in all the label in the ChIP-PCR data should be changed into "Enrichment".

2. In the manuscript, the authors used "human endometrial stromal cells (hESCs)" while the hESCs is usually short for "human Embryonic Stem Cells", I suggest to make a little change into "human endometrial stromal cells (HESCs)", which may reduce potential confusion.

*Reviewer #3 (Recommendations for the authors):*

1. The scientific background should be more clear and straightforward, it would be interesting that the authors discuss it in the manuscript.

a. The connection between receptivity and decidualization is confused. Decidualization is a differentiation process that regulates mainly the stromal compartment regulating embryo invasion, the next step of embryo adhesion, which is mainly govern by endometrial receptivity affecting the epithelial compartment. So, why the authors linked these processes and what is the connection?

b. Intro (lines 59-61): The receptive endometrium in human requires remodeling of stromal cells under the regulation of rising progesterone and intracellular cyclic AMP, which will undergo more extensive transformation driven by factors secreted from embryos. Which reference support this affirmation?

c. In the introduction ´Insufficient decidualization in endometrium is related to failed embryo implantation, unexplained infertility, recurrent spontaneous abortion (Coulam 2016), intrauterine growth retardation (Lefevre, Palin et al., 2011), and preeclampsia (Garrido-Gomez, Dominguez et al., 2017). However, the underlying molecular mechanism governing the endometrial decidualization remains enigmatic (Okada, Tsuzuki et al., 2018)´. There is lack of references to relate insufficient decidualization with failed embryo implantation and insufficient decidualization with unexplained infertility in this paragraph. The fact is that deficient decidualization relates to defects in embryo invasion and subsequent placentation and pregnancy complications. But the authors are trying to connect receptivity with insufficient decidualization and infertility. This is unclear and the authors need to be more precise that the scientific background that is exposed in this manuscript.

d. Intro: "both SOX4 and PGR have been demonstrated to be aberrantly downregulated in the endometrium of endometriosis (EMS) patients suffering from implantation failure". It is confused the focus of this study connecting role of SOX4 in decidualization with patients with implantation failure. If the authors want to relate SOX4 with implantation failure should demonstrate the connection with endometrial receptivity and embryo adhesion.

2. The clinical data included in this manuscript is very concise and need to be completed.

a. The authors described that some experiment was performed with normal samples, clinically what means ´normal´? The authors should define the clinical characteristics, the sample size and when the samples were collected.

b. Is the Table S2 (line 933) the equivalent to supplementary Table 1 (line 999)? please review it. This table include the clinical information about the patients analyzed in the last part of the article, but the authors should include another table including the clinical data of the patients that were recruited to collect the endometrial samples that are the source of primary hESCs used in the rest of paper. Also, this table include abbreviations that need to be defined and the history of previous pregnancies of controls to prove fertile status.

3. There are some general aspects that must be improved in the drafting of the results to improve the comprehensibility, much more details in some experiments are needed.

a. The nomenclature of cell types needs to be clearer, sometimes it is confusing to know whether you are using cell lines or primary endometrial stromal cells isolated from biopsies (e.g. In line 120 "endometrial stromal cells"), they are cell line or primary cells? it must be specified also in figure 1 legend, in line 129 "in endometrial biopsy samples obtained from healthy reproductive-aged volunteers" linked with Figure 1C, it would be better to specify which type of cells are you using in figure 1 because it seems that there are 2 type of cells. The authors explained results using immortalized cells and 293T cells, this are same cells? Please detail.

b. Another nomenclature issue to improve is the use of different terms for non-decidualized and decidualized cells "undifferentiated and differentiated stromal cells"; "undecidualized and decidualized"; "non-decidualized and decidualized". The authors should use unified nomenclature.

c. Authors expose results analyzing decidualized and non-decidualized cells, but the figures do not show decidualized vs non-decidualized instead of this they shown D0 and D4. Why do they not use decidualized and non-decidualized cells at same time of treatment (D4)? This comparison let us to know what is happening with cells under hormonal treatment compared with cells growing at the same time without decidualization inductors. The author should explain how they performed the experiments and why they decided to make this experimental design.

d. I have missed the times at which some treatments are performed, such as the decidualization treatments in Figure 2 A and the silencing and overexpression treatments. It should be specified for how long the silencing and overexpression is performed as well as decidualization. Additionally, in some treatments you must specify the dose of the drug you are using.

e. Why did you performed an RNA-Seq only 2 days after decidualization when in all of you experiments you show that decidualization markers increase their value from day four (e.g. Figure 3 B and C)? This may be somewhat confusing; the authors should explain this decision.

4. Some experimental part of the manuscript could be improved it:

a. Western Blot in general you have to specify in which day you have performed it, in some cases it has not been mentioned (e.g. Figure 3A). Also, the authors need to well-defined the double band that found in the PGR protein experiment, as well as, to improve some GAPDH western blot where the bands are undistinguished between conditions (example figure 2D and Figure 3G). Furthermore, graphic including the band quantification and statistical analysis in each western blot could help supporting the findings.

b. Immunostaining should be more described along the manuscript explaining that are based in the analysis of endometrial tissue and to specify details such as day of sample collection.

5. The authors need to improve details on methodology.

a. RNAseq data need to be explain in detail. Figure 1A-B: "(A) Expression of all transcription factors in human non-decidualized ESCs by RNA-Seq. (B) Expression of SOX family genes in human non-decidualized ESCs by RNA-Seq" is based in the normal endometrium stromal cells analysis and figure legend define that are non-decidualized cells. Which cells are analysed in RNAseq analysis? Please, do not use the term normal to clinically define type of sample. Which experimental design was used? Sample size (N)? Raw and normalized counts? Filtered and quality control process? Which statistics were used? Please improve the results showed for RNAseq analysis.

b. Versions of software and bioinformatic tools need to be included.

c. New sections in M and M need to be included for RNA extraction and quality evaluation (specifying RIN threshold consider), gene ontology enrichment, and real-time PCR.

d. How batch effect was evaluated, and variables tested for controlling confounding effects.

e. What function was used in edgeR for identify differentially expressed genes (e.g. exactTest), p-value adjustment and threshold for consider a transcript significantly deregulated, the fold change obtained and if there was some threshold to select genes to be included in subsequent analysis.

f. Antibody concentrations used in immunoprecipitation and immunostaining must be included. Also, reagent concentrations for transfection including lipofectamine and times followed to procced such as medium change during transfection. We suggest creating a Key Resource Table with all the chemical compounds, kits, antibodies, plasmids, and software used with the references and concentrations with the aim of being more transparent and increasing the reproducibility of experiments.

6. Regarding the discussion there are some aspects to be commented:

a. Line 409: The paper of SOX4 in decidualization linked with implantation failure in endometriosis could be improved. In this sense I suggest a proposal about how the authors think about the mechanism underlying with these two biological processes.

b. PGR isoforms A and B have different roles during the menstrual cycle, being the isoform B the predominant in decidualization. Authors should recognize it is a limitation not included this differentiation in the manuscript.

---

## [Author Response]

Some Essential Revisions that require your careful attention are listed below:– Expand Materials and methods section to provide more experimental details.

We have made the corresponding modification in the “Materials and methods” section in the revision as suggested.

– Provide Rationale for Sox4 ChIP-seq using immortalized cells and provide details of timing for ChIP-seq studies.

We conducted new ChIP-seq experiment using the primary endometrial stroma cells and got the similar results as in the immortalized cells (See revised Figure 4F) and the data have been deposited in the Gene Expression Omnibus database. The details of the timing are provided in the revised manuscript (See line 553-555).

– Provide data on knock down of SOX4 and HERC4 to rescue decidualization gene expression.

We conducted new experiments. The data on knock down of SOX4 and HERC4 to rescue decidualization gene expression is included in Figure 6—figure supplement 1F and also described in the revised manuscript (See line 297-298).

– Indicate unregulated genes in the absence of SOX4 in ChIP-seq experiments.

The data for upregulated genes in the absence of SOX4 is shown in the revised Supplementary Figure 4, and we also described this in the revised manuscript (See line 220-225).

– Clarify discussion on receptivity and decidualization.

We added the discussion on receptivity and decidualization in the revised manuscript (See line 54-61 and line 352-359).

– Provide more clinical data including sample size, characteristics and sample collection periods.

The detailed information for clinical data was provided in the Supplementary file 1. Clinical information of all patients and controls.

– Numerous methodological details need to be provided as suggested by Reviewer 3.

We provided the methodological details as reviewer suggested in the revised manuscript (See “Materials and methods” section).

Reviewer #1 (Recommendations for the authors):(1) An overarching issue is that the Results and Figure legends are not entirely descriptive about the experiments conducted to generate the data provided in the figures. The reader must delve into the Materials and methods to figure out how the experiment was conducted in terms of the approach. This issue can be rectified by providing more detail on the experimental approach for each major investigation and result. The Materials and methods need to be very clear on how experiments were conducted and analyzed.

Many thanks for this concern. We provided detailed information on the experimental approach for each major investigation and result. We also provided entire description in the figure legend.

(2) Line 188: What is the rationale for not using a SOX4 antibody to conduct the ChIP-seq studies in non-immortalized cells as opposed to overexpression of using a transfection approach? More detail needs to be presented on the treatments and timing of SOX4 ChIP-seq.

Indeed, we have tried commercially available antibodies as much as possible (Abcam, ab86809; Abcam, ab80261; Santa Cruz, sc-518016; Diagenode, C15310129). But none of them work well for ChIP-seq assay in the stromal cells, so we choose the exogenously expressed HA-Tagged SOX4 to explore the genome-wide binding sites of SOX4, and the exogenously expressed level was controlled at a comparable level to endogenous SOX4. Moreover, during the revision, the ChIP-seq experiment in the primary stromal cells recapitulated the results in non-immortalized cells (See revised Figure 4F), and the data have been deposited in the Gene Expression Omnibus database.

(3) Figures 6M and 6N: The figure legend indicates that this experiment was repeated three times, but the graphs do not provide error bars. Was the data for each Western blot quantified with respect to PGR protein and normalized to GAPDH? If so, why are error bars not present on the graphs?

Thanks for this concern. The statistical analysis of quantification of PGR expression was calculated in three independent experiments after normalized to GAPDH. In the revised figures, the error bars were present on the graphs.

(4) Supplemental Figure 4: The authors contend that HERC4 overexpression decreases PGR abundance, but the level of PGR protein is not assessed, and the PGR immunolocalization data is not very convincing.

Thanks for this concern. Besides the immunolocalization data, the level of PGR protein was also assessed by immunoblot and PGR abundance was decreased in the presence of HERC4 overexpression (See revised Figure 6K).

(5) Functional studies should be conducted to rescue decidualization gene expression with knockdown of HERC4 and SOX4. Alternatively, the authors can express a HERC4 resistant form of PGR and investigate the impact of SOX4 on decidualization.

Many thanks for this concern to strength our manuscript. We conducted new experiment to knockdown both HERC4 and SOX4 to explore the decidualization status and uncovered that HERC4 knockdown partially rescued IGFBP1 expression caused by the SOX4 deficiency (See revised Figure 6—figure supplement 1F).

(6) Figure 8: The data on EMS-3 are very intriguing but incomplete. If IGFBP1 is enhanced by SOX4 and PGR overexpression, then FOXO1 should also be investigated as well as PRL and other SOX4 and/or PGR responsive genes. In essence, the endometriosis angle for this patient is incomplete as presented.

As suggested, we conducted the new experiment to investigate FOXO1 expression when SOX4 and PGR were overexpressed, and found that FOXO1 was upregulated (See revised Figure 8K).

(7) The manuscript would benefit from editing to correct problems with English grammar and usage.

We corrected the problems with English grammar and usage.

(8) Line 356: HOX010 and HOXO11 should be HOXA and not HOXO.

Many thanks for this suggestion to strength our manuscript and we have made correction.

(9) Discussion: The fact that different patients with endometriosis-associated RIF have varying defects in several transcription factors should be highlighted in the Discussion. In essence, the underlying variability in several of the key transcription factors can all lead to the syndrome of progesterone resistance.

Thanks for this valuable concern, and we added the discussion for several transcription factors, such as factors in GATA and HOX family, which are reported to be associated endometriosis and progesterone resistance (See line 431-435).

Reviewer #2 (Recommendations for the authors):1) In Figure 1, the authors have examined the SOX4 expression during the menstrual cycle, how about its expression in the decidual tissue during the pregnancy?

Thanks for concerns to strengthen our manuscript. We detected the expression of SOX4 in the decidual tissue during the early pregnancy and found that the SOX4 was also highly expressed in the decidual cells during the pregnancy (See revised Figure 1C).

2) In Figure 4 The author showed the overlap the down-regulated genes in the absence of SOX with the ChIP data, how about the up-regulated genes?

We also provided the information for the up-regulated genes in the revised Supplementary Figure 4, and added the description in the revised manuscript (See line 220-225).

3) In Figure 4 the motif analysis for SOX4 ChIP-seq suggested the potential co-binding of regulatory region between the SOX4 and other proteins such as FOSL2, did the author tried to test the protein physical interaction during the decidualization?

Thanks for this thoughtful concern. We conducted the co-immunoprecipitation experiment as the reviewer suggested. The results showed that the there was no interaction between the SOX4 and FOSL2 in our co-IP system (See Author response image 1), suggesting SOX4 cannot strongly interact with the FOSL2.

**Author response image 1. sa2fig1:** HA antibody was used to immunoprecipitant HA-tagged SOX4 in stromal cells, the immunoprecipitated proteins was blotted with the indicated antibodies.

4) In Figure 5, while the proteasome inhibitor MG-132 could almost rescue the PR protein degradation in the SOX4 shRNA treatment, did the authors consider whether other protein degradation system, such as the lysosome mediated protein degradation may also be involved? This may deserve some discussions.

Thanks for this concern. We conducted the experiment and found that the inhibitor of lysosome cannot rescue the protein degradation (See revised Figure 5—figure supplement 1C and line 242-244).

Other suggestions:1. The "Eichment" in all the label in the ChIP-PCR data should be changed into "Enrichment".

Many thanks for this suggestion and we have corrected this mistake in the revised figures.

2. In the manuscript, the authors used "human endometrial stromal cells (hESCs)" while the hESCs is usually short for "human Embryonic Stem Cells", I suggest to make a little change into "human endometrial stromal cells (HESCs)", which may reduce potential confusion.

As suggested, we have corrected it.

Reviewer #3 (Recommendations for the authors):1. The scientific background should be more clear and straightforward, it would be interesting that the authors discuss it in the manuscript.a. The connection between receptivity and decidualization is confused. Decidualization is a differentiation process that regulates mainly the stromal compartment regulating embryo invasion, the next step of embryo adhesion, which is mainly govern by endometrial receptivity affecting the epithelial compartment. So, why the authors linked these processes and what is the connection?

Different from the scenario in the rodent models in which the decidualization occurred after the embryo attachment, decidualization of the human endometrium does not require embryo adhesion. The human endometrial stromal compartment can initiate the decidualize during the progesterone-dominant early secretory phase of menstrual cycle. When the endometrium entered into the receptive state mainly dominating by the endocrine progesterone and estrogen, the stromal compartment has initiated the decidualization, which has also been demonstrated by the recent single cell RNA-Seq data (Single-cell transcriptomic atlas of the human endometrium during the menstrual cycle. Nat Med. 2020 Oct;26(10):1644-1653. PMID: 32929266). The implantation window open, which accompanied with the receptive endometrium, accompanied with a widespread decidualization feature in the stromal fibroblasts, which was confirmed by our data about the decidualization marker IGFBP1 immunostaining (Figure 8I). Just as the reviewer stated, the embryo adhesion occurred between the blastocyst and the endometrial epithelium, but the proper differentiation of epithelium preparing for adhesion with the embryo is also regulated the stromal compartment, which have been demonstrated by mouse models. Only when the epithelium and stroma compartment properly differentiated under the influence of endocrine progesterone and estrogen, the endometrial receptivity can be established well, and the differentiation process for stromal compartment is the decidualization. In this study, we focus the decidualization, which means the stromal cell differentiation, directed by the progestogen signal for preparing the endometrial receptivity.

b. Intro (lines 59-61): The receptive endometrium in human requires remodeling of stromal cells under the regulation of rising progesterone and intracellular cyclic AMP, which will undergo more extensive transformation driven by factors secreted from embryos. Which reference support this affirmation?

Thanks for this concern. We have provided references for this description and reconstruct this sentence:

“The receptive endometrium in human requires remodeling of stromal cells under the regulation of rising progesterone and intracellular cyclic AMP, which will undergo more extensive transformation after embryo implantation”.

c. In the introduction ´Insufficient decidualization in endometrium is related to failed embryo implantation, unexplained infertility, recurrent spontaneous abortion (Coulam 2016), intrauterine growth retardation (Lefevre, Palin et al., 2011), and preeclampsia (Garrido-Gomez, Dominguez et al., 2017). However, the underlying molecular mechanism governing the endometrial decidualization remains enigmatic (Okada, Tsuzuki et al., 2018)´. There is lack of references to relate insufficient decidualization with failed embryo implantation and insufficient decidualization with unexplained infertility in this paragraph. The fact is that deficient decidualization relates to defects in embryo invasion and subsequent placentation and pregnancy complications. But the authors are trying to connect receptivity with insufficient decidualization and infertility. This is unclear and the authors need to be more precise that the scientific background that is exposed in this manuscript.

Thanks very much for this suggestion. We have added the reference demonstrating the relationship between the decidualization and uterine receptivity (See line 62). Moreover, we also provided the scientific background for relating insufficient decidualization with failed embryo implantation (See line 352-359 in “Discussion” section).

d. Intro: "both SOX4 and PGR have been demonstrated to be aberrantly downregulated in the endometrium of endometriosis (EMS) patients suffering from implantation failure". It is confused the focus of this study connecting role of SOX4 in decidualization with patients with implantation failure. If the authors want to relate SOX4 with implantation failure should demonstrate the connection with endometrial receptivity and embryo adhesion.

Thanks very much for this suggestion. The similar concern for the relation between the decidualization and endometrial receptivity. We have added some discussion for the decidualization and uterine receptivity, including the embryo adhesion (See line 352-359).

2. The clinical data included in this manuscript is very concise and need to be completed.a. The authors described that some experiment was performed with normal samples, clinically what means ´normal´? The authors should define the clinical characteristics, the sample size and when the samples were collected.

Thanks for this concern. The normal samples mean that the endometrial samples were collected from the women who have given birth, and we have changed the ´normal ´with ´healthy´. We also provided a table about clinical information for the sample collection (See Supplementary file 1. Clinical information of all patients and controls).

b. Is the Table S2 (line 933) the equivalent to supplementary Table 1 (line 999)? please review it. This table include the clinical information about the patients analyzed in the last part of the article, but the authors should include another table including the clinical data of the patients that were recruited to collect the endometrial samples that are the source of primary hESCs used in the rest of paper. Also, this table include abbreviations that need to be defined and the history of previous pregnancies of controls to prove fertile status.

Thanks for this concern to strength our manuscript. We have provided the new Supplementary file 1. Clinical information of all patients and controls. for the clinical sample which was used for isolating the primary cultured cell.

3. There are some general aspects that must be improved in the drafting of the results to improve the comprehensibility, much more details in some experiments are needed.a. The nomenclature of cell types needs to be clearer, sometimes it is confusing to know whether you are using cell lines or primary endometrial stromal cells isolated from biopsies (e.g. In line 120 "endometrial stromal cells"), they are cell line or primary cells? it must be specified also in figure 1 legend, in line 129 "in endometrial biopsy samples obtained from healthy reproductive-aged volunteers" linked with Figure 1C, it would be better to specify which type of cells are you using in figure 1 because it seems that there are 2 type of cells. The authors explained results using immortalized cells and 293T cells, this are same cells? Please detail.

Thanks for this suggestion. We have added the detailed information in the revised manuscript and figure legends. 293T cells was utilized in this section mainly due to lower transfection efficiency in immortalized stromal cells. Similar results in both cell types also corroborate the regulation of PR protein stability by the ubiquitin E3 ligase HERC4.

b. Another nomenclature issue to improve is the use of different terms for non-decidualized and decidualized cells "undifferentiated and differentiated stromal cells"; "undecidualized and decidualized"; "non-decidualized and decidualized". The authors should use unified nomenclature.

Thanks for this concern to strength our manuscript and we have uniformly changed into the “undecidualized and decidualized".

c. Authors expose results analyzing decidualized and non-decidualized cells, but the figures do not show decidualized vs non-decidualized instead of this they shown D0 and D4. Why do they not use decidualized and non-decidualized cells at same time of treatment (D4)? This comparison let us to know what is happening with cells under hormonal treatment compared with cells growing at the same time without decidualization inductors. The author should explain how they performed the experiments and why they decided to make this experimental design.

The main purpose of this experiment is to determine the gene expression during the decidualization process, that is from the beginning of treatment (D0) throughout to D6. The similar approach has been used in the published paper (Recurrent pregnancy loss is associated with a pro-senescent decidual response during the peri-implantation window. Commun Biol. 2020 Jan 21;3(1):37. PMID: 31965050).

d. I have missed the times at which some treatments are performed, such as the decidualization treatments in Figure 2 A and the silencing and overexpression treatments. It should be specified for how long the silencing and overexpression is performed as well as decidualization. Additionally, in some treatments you must specify the dose of the drug you are using.

Thanks for this concern to make our description clearer. HESCs were cultured with differentiation medium for two days, since the expression of PR increased significantly on the second day of HESC differentiation. For the knockdown and overexpression experiment, the silencing and overexpression was first performed without differentiation stimulus, 24 hours later hESCs were decidualization for another 2 days. This information had been added in the revised manuscript (See line 546-549). The information including the doses of drug have been included in the Key resource table.

e. Why did you performed an RNA-Seq only 2 days after decidualization when in all of you experiments you show that decidualization markers increase their value from day four (e.g. Figure 3 B and C)? This may be somewhat confusing; the authors should explain this decision.

Thanks for this concern. During the process of decidualization, SOX4 was significantly induced as early as 2 days after decidualization, and the expression of decidualization markers PRL and IGFBP1 was obvious increased at the later time, such on D4 and D6. To explore the direct effect of SOX4 on the regulation for decidualization, the time when SOX4 was induced was chose for the RNA-Seq to minimize the secondary effects. The decidualization status as determined by the marker genes expression was on D4, when the marker gene expression increased to a high level.

4. Some experimental part of the manuscript could be improved it:a. Western Blot in general you have to specify in which day you have performed it, in some cases it has not been mentioned (e.g. Figure 3A). Also, the authors need to well-defined the double band that found in the PGR protein experiment, as well as, to improve some GAPDH western blot where the bands are undistinguished between conditions (example figure 2D and Figure 3G). Furthermore, graphic including the band quantification and statistical analysis in each western blot could help supporting the findings.

Thanks for this concern. We added the information about the timing in the revised figure legend. The double band in the WB data for PR indicate PRA and PRB respectively and we have added this information in the revised manuscript (See line 235). As suggested by the reviewer, we improved the bands of GAPDH by rerunning the WB and also provided the data for band quantification and statistical analysis (See revised Figure 2 and 3).

b. Immunostaining should be more described along the manuscript explaining that are based in the analysis of endometrial tissue and to specify details such as day of sample collection.

Thanks for this concern. The detailed information for Immunostaining has been provided as reviewer suggested in the revised manuscript (See line 618-621).

5. The authors need to improve details on methodology.a. RNAseq data need to be explain in detail. Figure 1A-B: "(A) Expression of all transcription factors in human non-decidualized ESCs by RNA-Seq. (B) Expression of SOX family genes in human non-decidualized ESCs by RNA-Seq" is based in the normal endometrium stromal cells analysis and figure legend define that are non-decidualized cells. Which cells are analysed in RNAseq analysis? Please, do not use the term normal to clinically define type of sample. Which experimental design was used? Sample size (N)? Raw and normalized counts? Filtered and quality control process? Which statistics were used? Please improve the results showed for RNAseq analysis.

Thanks very much for the reviewer’s suggestion. The term “normal” was changed to “non-decidualized ESCs” cell in the revised manuscript. RNA-Seq was perform in non-decidualized ESCs for Figure 1A-B. The average FPKM values (N = 3) of all transcription factors were used for these plots. Quality scores of reads above 30 was kept for alignment and only uniquely mapped reads were used for downstream quantification analysis by exact negative binomial test in edgeR package of R. The results of RNA-Seq of Figure 1A and 1B were improved per suggestion.

b. Versions of software and bioinformatic tools need to be included.

The softwares for RNA-Seq and ChIP-seq analysis including STAR (2.7.3a) for alignment, MACS2 (2.2.7.1) for peakcall of ChIP-seq, ggplot2 (3.3.5) for visualization, ngs.plot.r (2.61) for peak heatmap of ChIP-seq and the packages of edgeR (3.9), Complexheatmap (2.4.3), ChIPseeker (1.24.0) in R (4.1).

c. New sections in M and M need to be included for RNA extraction and quality evaluation (specifying RIN threshold consider), gene ontology enrichment, and real-time PCR.

The description about RNA extraction and quality evaluation, gene ontology enrichment, and real-time PCR had been modified or added as reviewer’s suggestion.

d. How batch effect was evaluated, and variables tested for controlling confounding effects.

To reduce the batch effect, the libraries of all RNA samples were prepared at the same time. The pooled RNA libraries were sequenced in a same flow cell. After sequencing, the sequencing depth and principal component analysis of all samples were evaluated. After library size calculation, sample dispersion evaluation and normalization, gene expression was calculated as FPKM (Fragments per kilobase of transcript per million mapped reads).

e. What function was used in edgeR for identify differentially expressed genes (e.g. exactTest), p-value adjustment and threshold for consider a transcript significantly deregulated, the fold change obtained and if there was some threshold to select genes to be included in subsequent analysis.

The function of exactTest in edgeR was applied for p-value calculation. Fold changed above 2 or less than 0.5 and p-Value < 0.05 were the significance threshold for consideration of differential expression. Only those differentially expressed genes (DEG) met the significance threshold were selected for downstream analysis.

f. Antibody concentrations used in immunoprecipitation and immunostaining must be included. Also, reagent concentrations for transfection including lipofectamine and times followed to procced such as medium change during transfection. We suggest creating a Key Resource Table with all the chemical compounds, kits, antibodies, plasmids, and software used with the references and concentrations with the aim of being more transparent and increasing the reproducibility of experiments.

Thanks for this concern. We have provided more tables for all the chemical compounds, kits, antibodies, and plasmids (See Key resources table and Supplementary file 3. The plasmids used in this study).

6. Regarding the discussion there are some aspects to be commented:a. Line 409: The paper of SOX4 in decidualization linked with implantation failure in endometriosis could be improved. In this sense I suggest a proposal about how the authors think about the mechanism underlying with these two biological processes.

Thanks very much for this concern. We narrow down the statement about the connection between decidualization and implantation to prevent the disruption the focus of this investigation.

b. PGR isoforms A and B have different roles during the menstrual cycle, being the isoform B the predominant in decidualization. Authors should recognize it is a limitation not included this differentiation in the manuscript.

Thanks for this concern. As suggested by the reviewer, both PRA and PRB were expressed in the stromal cells, and PRB played predominant role in decidualization, we also observe the expression of PRA and PRB were both influenced by the SOX4 and since the PRB was more critical for decidualization, the regulation of HERC4 on PR was mainly focused on PRB.